# DECOUPLE QUANTIZATION STEP AND OUTLIER-MIGRATED RECONSTRUCTION FOR PTQ

## ABSTRACT

Post-training quantization (PTQ) is a popular technique for compressing deep learning models due to its low cost and high efficiency. However, in some extremely low-bit settings, PTQ still suffers from significant performance degradation. In this work, we reveal two related obstacles: (1) the setting of weight's quantization step has not been fully explored, and (2) the outlier activation beyond clipping range are ignored in most methods, which is especially important for lightweight models and low-bit settings. To overcome these two obstacles, we propose **DOMR**, to (1) fully explore the setting of weight's quantization step into five cases through **D**ecoupling, based on the ignored fact that integer weight (different from integer activation) can be obtained early before actual inference deployment, (2) save outliers into the safe clipping range under predefined bitwidth with **O**utlier-**M**igrated **R**econstruction, based on the nature of CNN structure and PTQ's clipping operation. More outliers saved equals to breaking the bitwidth shackle of a predefined hardware thus brings better performance. Extensive experiments on various networks demonstrate that DOMR establishes a new SOTA in PTQ. Specifically, DOMR outperforms the current best method by 12.93% in Top-1 accuracy for W2A2 on MobileNet-v2. The code will be released.

## 1 INTRODUCTION

Deep neural networks' huge computational cost has posed a challenge to their deployment in real-world applications. To solve this problem, various model compression techniques (Han et al., 2015; Hinton et al., 2015) have been studied. Low-bit model quantization is one of the commonly used methods, which generally consists of Quantization-Aware Training (QAT) and Post-Training Quantization (PTQ). PTQ only needs a tiny amount of unlabeled data and does not demand the full training pipeline. Therefore, PTQ is always the first choice for fast model quantization. It is a common PTQ method (Krishnamoorthi, 2018) to search the quantization step based on the MSE error between FP32 and quantizied values. However, in lower bits like 4 bits or 2 bits, traditional PTQs suffer significant accuracy degradation. Some recent algorithms proposed to reconstruct quantized output features from two aspects: (a) Reconstruction granularity: AdaRound (Nagel et al., 2020b) proposed a layer-by-layer reconstruction method, and introduce adaptive rounding parameters for weights. BRECQ (Li et al., 2021b) found out reconstructing block-by-block performs better. NWQ (Wang et al., 2022) further proposed a network-wise PTQ. (b) Reconstruction smoothness: QDROP (Wei et al., 2022) proposed to randomly quantize a part of a tensor like dropout. MRECG (Ma et al., 2023) and Bit-Shrink (Lin et al., 2023) solve the oscillation and sharpness problems in PTQ respectively.

These techniques boost PTQ performance. However, on one hand, during PTQ feature reconstruction above, the quantizer's step (quant-step) of weight in these methods are frozen after initialization. What will we gain if weight's quant-step is cooperatively optimized with other parameters like weight's adaptive rounding parameter and activation's quant-step? On the other hand, current PTQ methods inherited treat weight's quant-step $s_w$ and activation's quant-step $s_a$ equally while ignoring the fact integer weight can be obtained early before inference while integer activation has to be computed during inference, as described in section 3.1. This means after computing the integer weight using $s_w$ during fake quantization simulation, the $s_w$ used to convert integer weight into the FP32 counterpart can be different. As shown in formula 4. what will we gain if we decouple the original single quant-step of weight $s_w$ into a quant-step $s_w$ and a de-quantizer's step (dequant-step) $s_w'$ according to their different function? As far as we know, these two problems above has not been fully

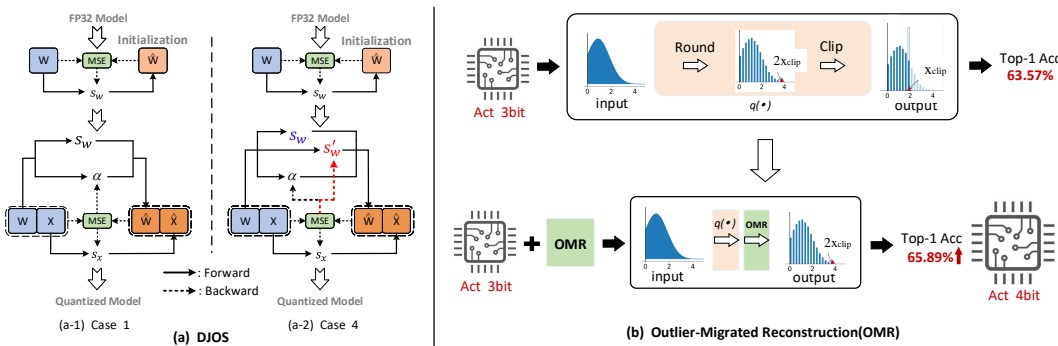

Figure 1: (a): Fully explore weight's quant-step into five cases with DJOS, details in Sec 3.1. Case 1 represents traditional PTQ's quant-step: single and not learnable. Case 4 represents DJOS: quant-step $S_w$ is decoupled into a frozen quant-step $S_w$ and learnable dequant-step $S'_w$. (b): Break Bitwidth Shackle with Outlier-Migrated Reconstruction (OMR), details in Figure 3. The top of (b) describes a classic PTQ process in W3A3, denoted by $q(\cdot)$. With OMR equipped after $q(\cdot)$ as the bottom of (b), more outliers in $(X_{clip}, 2X_{clip}]$ can be saved under predefined bitwidth. Thus 3-bit activation in OMR-solvable structures can be expanded to 4 bits equally.

explored yet. According to whether the initialized quant-step of weight is decoupled and learnable or not during quantized feature reconstruction, we can present them into five cases as described in Sec 3.1, where Case (1) is the current setting of PTQ. We experimentally find out Case (4), where we **D**ecouple quant-step, then **J**ointly **O**ptimize dequant-**S**tep only (**DJOS**), consistently provides the best performance. DJOS outperforms current SOTA without any other extra computation or parameters during inference. It can also be an orthogonal plugin for other classic PTQ methods.

Besides, in current PTQs, the outlier activation beyond clipping range will be truncated due to PTQ's clipping operation. However, these outliers are important, especially for lightweight models and extremely low bit settings as described in Figure 2. The simplest way to cover more outliers is to enlarge quantization bitwidth, however, which is impossible under a predefined bitwidth accelerator. Therefore, we try to preserve these important outliers which should have been clipped under a predefined bitwidth. Based on the nature of CNN struc-

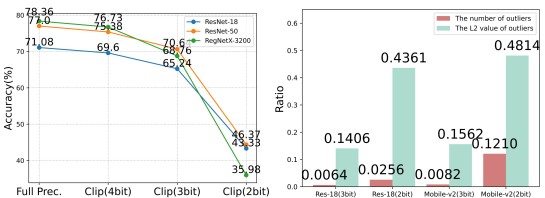

Figure 2: left: Acc@1 on ImageNet for fp32 and clipping-only quantized networks. right: The ratio and importance (L2 value) of outliers.

ture and the clipping operation, we propose to migrate the outliers down into the clipping range and store them in some newly-built channels, so they can get through the clipping gate safely. Then, with a mathematically equivalent transformation, we make some special modifications on corresponding weights. Thus a functionally-identical network is caused without retraining except for more outlier activation being preserved. In such a way, namely **O**utlier-**M**igrated **R**econstruction(**OMR**), we successfully achieve the target that more outliers can be reconstructed under predefined bitwidth. It is equal to breaking bitwidth shackle of hardware, since the outliers in $(X_{clip}, 2X_{clip}]$ should have been preserved by designing one more bit on hardware as shown on the right (b) of Figure 1. The fair comparison with the same FLOPs between OMR-quantized network and channels-increased NWQ-quantized network demonstrates the effectiveness of our OMR. Therefore, though our OMR bring extra FLOPs, it is still worthy under a predefined bitwidth hardware. Our main contributions on PTQ are summarized as follows:

- For weight PTQ, we propose DJOS. As far as we know, we are the first to fully investigate different settings of weight's quant-step into five cases. We find Case (4), where we **D**ecouple the single quant-step of weight into a quant-step and a dequant-step, then **J**ointly **O**ptimize dequant-**S**tep only (**DJOS**), consistently provides the best performance. DJOS outperforms other PTQ methods and is orthogonal to other PTQs, without any other extra computation or params in deployment.

- For activation PTQ, we propose OMR, based on the nature of CNN structure and PTQ's clipping operation. OMR saves the outlier activation beyond clipping range while preserves the precision of inner values. It is equal to breaking the bitwidth shackle of hardware, thus $b$-bit activation in OMR-solvable structures can be expanded to $(b + 1)$ bits equally. OMR can be easily inserted as a plug-and-play module into current PTQ libraries to further improve performance orthogonally.

- We evaluate DJOS+OMR$\Rightarrow$DOMR on classic computer vision tasks across various CNN models and ViTs. Experimental results prove that our DOMR sets up a new SOTA for PTQ.

## 2 RELATED WORK

PTQ takes in a well-trained 32 bit floating-point (FP32) model then covert it into a low-bit fixed-point counterpart directly. For weight quantization, Adaround (Nagel et al., 2020a) found that the commonly used rounding-to-nearest operation will be sub-optimal. Thus it proposed an adaptive rounding for weight. BRECQ (Li et al., 2021a) found out block-by-block reconstruction behaves better than the former layer-by-layer ones. NWQ (Wang et al., 2022) further proposed a network-wise PTQ by fully leveraging inter-layer dependency. QDROP (Wei et al., 2022) proposed to jointly optimize quantization error of both weight and activation. Meanwhile, it proposed to randomly quantize only a part of a tensor like dropout. MRECG (Ma et al., 2023) tried to Solve Oscillation problem in PTQ through a theoretical perspective. PD-Quant (Liu et al., 2023) proposed to consider global information based on prediction difference metric. Bit-Shrink (Lin et al., 2023) proposed to limit instantaneous sharpness for improving PTQ. PTQ4ViT (Yuan et al., 2022) and APQ-ViT (Ding et al., 2022) are proposed to solve the PTQ for Vision Transformer(ViT).

Different from all methods above ignoring the impact of weight's quant-step, we fully explore the different settings of weight's quant-step into five cases, and find the best case for PTQ.

In addition, the existing clipping operation permanently truncates the outlier activation beyond the threshold. However, these outliers might be crucial to the final task loss, especially for some lightweight networks. To alleviate this problem, OCS (Zhao et al., 2019) proposed "outlier channel splitting" to duplicate channels containing outliers then halve the channel values, which caused a functionally identical network and the affected outliers are moved to the center of distribution. Nevertheless, they mentioned that OCS was good at weight quantization but failed to quantize activation.

Different from OCS, in this paper, we aim at activation quantization. Instead of OCS's halving operation, we propose to migrate outliers, thus the outlier activation beyond the threshold can get through the clipping gate safely, meanwhile the original information in the range will not be squeezed narrower like OCS either. So it does a much better job on activation quantization than OCS. Such a functionally-identical modified network achieves one-bit higher representation in theory without modification on predefined hardware. It conveniently brings better quantized-feature reconstruction.

## 3 PROPOSED METHOD

**Preliminaries of PTQ:** A classic weight PTQ is shown as formula 1. $s_w$ is the quant-step of weights $w$. $w_l/w_u$ are the lower/upper bound of quantization levels. $\lfloor \cdot \rceil$ indicates rounding operation.

$$\hat{\boldsymbol{w}} = clip(\lfloor \frac{\boldsymbol{w}}{s_w} \rceil; w_l, w_u) \cdot s_w, \quad \min_{\hat{\boldsymbol{w}}} ||\hat{\boldsymbol{w}} - \boldsymbol{w}||_F^2 \tag{1}$$

Recent works aimed to improve the performance of PTQ by reconstructing quantized feature per layer/block/network. However, almost all these methods ignore the difference of how integer weight and integer activation are produced during practical inference. In addition, due to the clipping operation in quantization, the methods above failed to reconstruct the outlier activation beyond clipping threshold. Considering the two problems above, we propose DJOS and OMR as followed.

### 3.1 DECOUPLE AND JOINTLY OPTIMIZE QUANT-STEPS

**The current SOTA of PTQ**. MRECG (Ma et al., 2023)/NWQ (Wang et al., 2022) quantizes weight $w$ and activation $x$ as equation 2, where $h(\boldsymbol{\alpha})$ is the AdaRound (Nagel et al., 2020a) parameter of weight, $s_w(s_x)$ is the quant-step of weight(activation), $w_l(x_l)/w_u(x_u)$ is the lower/upper bound

of weight(activation). The initialized $s_w$ is obtained by minimizing the MSE between FP32 and quantized weight. Then $s_w$ is frozen when optimizing AdaRound $h(\boldsymbol{\alpha})$ and activation quantization.

$$\hat{\boldsymbol{w}} = clip(\lfloor \frac{\boldsymbol{w}}{s_w} \rfloor + h(\boldsymbol{\alpha}); w_l, w_u) \cdot s_w, \quad \hat{\boldsymbol{x}} = clip(\lfloor \frac{\boldsymbol{x}}{s_x} \rceil; x_l, x_u) \cdot s_x \qquad (2)$$

**Fake and Real Quantization**. As shown in formula 3, in order to better optimize PTQ on GPU, the quantization function has to be simulated with FP32, denoted as the left 'Fake Quantize' bracket. For practical inference acceleration, the FP32 simulated computation is transformed to be integer-arithmetic-only (Jacob et al., 2018), denoted as the middle 'Real Quantize' bracket.

$$\boldsymbol{y} = \underbrace{\sum \boldsymbol{wx}}_{FP32} \approx \underbrace{\sum \hat{\boldsymbol{w}}\hat{\boldsymbol{x}}}_{Fake\ Quantize} = \underbrace{s_w s_x \cdot \sum \boldsymbol{w_{int}} \left[ \frac{\boldsymbol{x}}{s_x} \right]}_{Real\ Quantize}, \boldsymbol{w_{int}} = clip(\lfloor \frac{\boldsymbol{w}}{s_w} \rfloor + h(\boldsymbol{\alpha}); w_l, w_u) \quad (3)$$

where $\lceil \cdot \rceil$ denotes rounding and clipping operations. $\boldsymbol{w_{int}}$ denotes integer weight, which can be obtained early before practical inference deployment as the right part of formula 3.

**Decouple quant-step of weight**. As shown in formula 3, for real quantization, the integer weight $\boldsymbol{w_{int}}$ and $\boldsymbol{s_w}$ are determined prior to deployment, and they can be treated as independent parameters. Given this property, for fake quantization, we propose to decouple the quant-step of weight into quant-step $s_w$, which quantize FP32 weight to integer value, and dequant-step $s'_w$, which de-quantize integer weight back to FP32 value. Thus formula 2 can be transformed to formula 4:

$$\hat{\boldsymbol{w}} = clip(\lfloor \frac{\boldsymbol{w}}{s_w} \rfloor + h(\boldsymbol{\alpha}); w_l, w_u) \cdot s_w \quad \Rightarrow \quad \hat{\boldsymbol{w}} = clip(\lfloor \frac{\boldsymbol{w}}{s_w} \rfloor + h(\boldsymbol{\alpha}); w_l, w_u) \cdot s'_w \qquad (4)$$

Correspondingly, the real quantization process is transformed as the following formulas:

$$\boldsymbol{y} = s'_w s_x \cdot \sum \boldsymbol{w_{int}} \left[ \frac{\boldsymbol{x}}{s_x} \right], \quad \boldsymbol{w_{int}} = clip(\lfloor \frac{\boldsymbol{w}}{s_w} \rfloor + h(\boldsymbol{\alpha}); w_l, w_u) \qquad (5)$$

**Jointly Optimize Quant-Steps**. Under the condition where the quant-step of weight can be decoupled, for the first time, we fully explore different settings of weight's quant-step into five cases, based on whether quant-step $s_w$ and de-quant step $s'_w$ are learnable or not after initialization.

**Case 1**: the original single quant-step $s_w$ is not decoupled as convention, and do not participate joint optimization during feature reconstruction after initialization, which is what current PTQ methods adopt, as shown on the left (a-1) of Figure 1.

**Case 2**: the original single quant-step $s_w$ is not decoupled as convention, and participates joint optimization during feature reconstruction.

**Case 3**: the original single quant-step $s_w$ is decoupled as $s_w$ and $s'_w$. Only quant-step $s_w$ participates joint optimization during feature reconstruction.

Table 1: Acc@1 on ImageNet among different quant-step settings across various nets.

| Methods | W/A | Mobile-v2 | Res-18 | Mnas2.0 |
|---|---|---|---|---|
| Case 1 (current PTQ) | 3/2 | 38.92 | 60.82 | 52.17 |
| Case 2 | 3/2 | 39.65 | 60.26 | 49.78 |
| Case 3 | 3/2 | 38.77 | 59.90 | 48.40 |
| **Case 4** | 3/2 | **42.60** | **61.06** | **54.19** |
| Case 5 | 3/2 | 41.42 | 60.86 | 49.33 |

**Case 4**: the original single quant-step $s_w$ is decoupled as $s_w$ and $s'_w$, and only dequant-step $s'_w$ participates joint optimization during feature reconstruction, as the left (a-2) of Figure 1.

**Case 5**: the original single quant-step $s_w$ is decoupled as $s_w$ and $s'_w$, and both participate joint optimization during feature reconstruction.

To evaluate their efficiency, we conduct experiments on MobileNet-v2, ResNet-18, and MnasNet2.0. The results are shown in Table 1.

We find out that Case 4, where we decouple the original quant-step $s_w$ as $s_w$ and $s'_w$ then make only dequant-step $s'_w$ participates joint optimization during feature reconstruction, as shown on the left (a-2) of Figure 1, consistently provides the best performance, which even does a better job than Case 5. We attribute the reason to the non-differentiable floor operation followed by quant-step $s_w$. To fluent gradient flow, we have to adopt Straight-Through Estimator(STE) (Bengio et al., 2013). However, PTQ's shortage of labeled training set and STE's mismatched gradient, whose approximation is indeed a little far way from the floor operation, make the optimization of $s_w$ extremely difficult. Therefore, it will achieve better performance to make the dequant-step $s'_w$ learnable only.

Through full exploration of these five cases, we propose to decouple the quant-step and dequant-step of weight into $s_w$ and $s_w'$, then freeze quant-step $s_w$ after initialization and embrace a learnable dequant-step $s_w'$ for joint optimization. The adopted joint optimization can be expressed as:

$$\min_{s_w', \boldsymbol{\alpha}, s_x} ||\hat{\boldsymbol{W}}\hat{\boldsymbol{x}} - \boldsymbol{W}\boldsymbol{x}||_F^2 \tag{6}$$

## 3.2 BREAK BIT SHACKLE WITH OUTLIER-MIGRATED RECONSTRUCTION

Among current PTQ methods, the activation beyond the clipping range will be truncated due to the clipping operation in quantization, which causes permanent information loss. As shown on the left of Figure 2, we use clipping operation only (no rounding operation) during quantization to analyze how much loss is caused when we directly clip outlier activation. As bitwidth goes lower, the lost clipped outliers make model performance drop sharply. Although the outliers account for only a small proportion in quantity, as shown on the right of Figure 2, they account for a large proportion in quality (denoted by L2 value: $||\boldsymbol{x}_{ij}||^2$).

We argue that if these high-quality outlier activation are saved properly, the PTQ performance will be improved greatly. However, we can not save more outliers while keep the same fine-grained quant-step simultaneously under a predefined bitwidth accelerator. Instead of ignoring this contradiction as before, we propose OMR to preserve more outlier activation. The cascade $[Conv + ReLU + Conv]$ is one of the most common structure in CNN, as shown at (a) of Figure 3. Since $BN$ layer is generally merged into its last Conv layer in PTQ, cascade $[Conv + BN + ReLU + Conv]$ can also be regarded as $[Conv + ReLU + Conv]$. Such a cascade can be denoted as follows:

$$\boldsymbol{y}^l = \boldsymbol{W}^l\boldsymbol{x}^l + \boldsymbol{b}^l, \quad \boldsymbol{x}^{l+1} = f(\boldsymbol{y}^l), \quad \boldsymbol{y}^{l+1} = \boldsymbol{W}^{l+1}\boldsymbol{x}^{l+1} + \boldsymbol{b}^{l+1} \tag{7}$$

The intermediate output $\boldsymbol{x}^{l+1}$ can be quantized into $\hat{\boldsymbol{x}}^{l+1}$ using the left of formula 8. Specifically, when $f(\cdot) = ReLU$, the left quantization formula is equivalent to the right of formula 8.

$$\hat{\boldsymbol{x}}^{l+1} = clip(\lfloor\frac{f(\boldsymbol{y}^l)}{s_x}\rceil; x_l, x_u) \cdot s_x, \quad \hat{\boldsymbol{x}}^{l+1} = clip(\lfloor\frac{\boldsymbol{y}^l}{s_x}\rceil; 0, x_u) \cdot s_x \tag{8}$$

Owing to the clipping operation, the outlier features, greater than $x_{clip} = x_u \cdot s_x$, in $\boldsymbol{x}^{l+1}$, are clipped to $x_{clip}$, which will cause this outlier part of features unable to be reconstructed or adjusted during PTQ process. Consequently, the final quantized model suffers huge performance drop.

The upper bound of quantized results $x_u = s_x \cdot (2^b - 1)$ is determined by quant-step $s_x$ and predefined bitwidth $b$. If we want to enlarge $x_u$ and cover more outliers, one way is to enlarge quant-step. However, a larger, or coarse-grained, quant-step will lead to larger discretization error for a converged quantized model. The other way is to enlarge bitwidth, which is impossible for a predefined-bitwidth accelerator. Is there a solution covering more outliers, while requiring the same bit?

Based on the commonly used structure as shown at (a) of Figure 3, called as OMR-solvable structures, if we can afford a little more calculation, the quantization level can be enlarged safely, meanwhile, the quantization bit can be kept as the same. Given that we want to enlarge the upper bound from $x_{clip}$ to $\beta x_{clip}$, $\beta \geq 1$, the activation quantization is correspondingly transformed to:

$$\hat{\boldsymbol{x}}^{l+1}_{[0,\beta x_{clip}]} = clip(\lfloor\frac{\boldsymbol{y}^l}{s_x}\rceil; 0, x_u) \cdot s_x + ... + clip(\lfloor\frac{\boldsymbol{y}^l - (\beta - 1) \cdot x_{clip}}{s_x}\rceil; 0, x_u) \cdot s_x \tag{9}$$

From the formula above, we can see the outlier activation is migrated back to the original quantization range successfully. Such a migration on feature can be compensated through some number (related to $\beta$) of channels added and some simple modification on weight and bias parameters. To simplify denotation, here we set $\beta = 2$. Thus the formula equation 9 can be simplified as:

$$\hat{\boldsymbol{x}}^{l+1}_{[0,2x_{clip}]} = clip(\lfloor\frac{\boldsymbol{y}^l}{s_x}\rceil; 0, x_u) \cdot s_x + clip(\lfloor\frac{\boldsymbol{y}^l - x_{clip}}{s_x}\rceil; 0, x_u) \cdot s_x \tag{10}$$

From the formula above, we first need to duplicate activation $\boldsymbol{y}^l$, translate the copied one down by $x_{clip}$ and concatenate them together as follows:

$$\boldsymbol{y}^{l'}_j = \begin{cases} \boldsymbol{y}^l_j & \text{if} \quad 0 \leq j < c_{out} \\ \boldsymbol{y}^l_{j-c_{out}} - x_{clip} & \text{if} \quad c_{out} \leq j < 2c_{out} \end{cases} \tag{11}$$

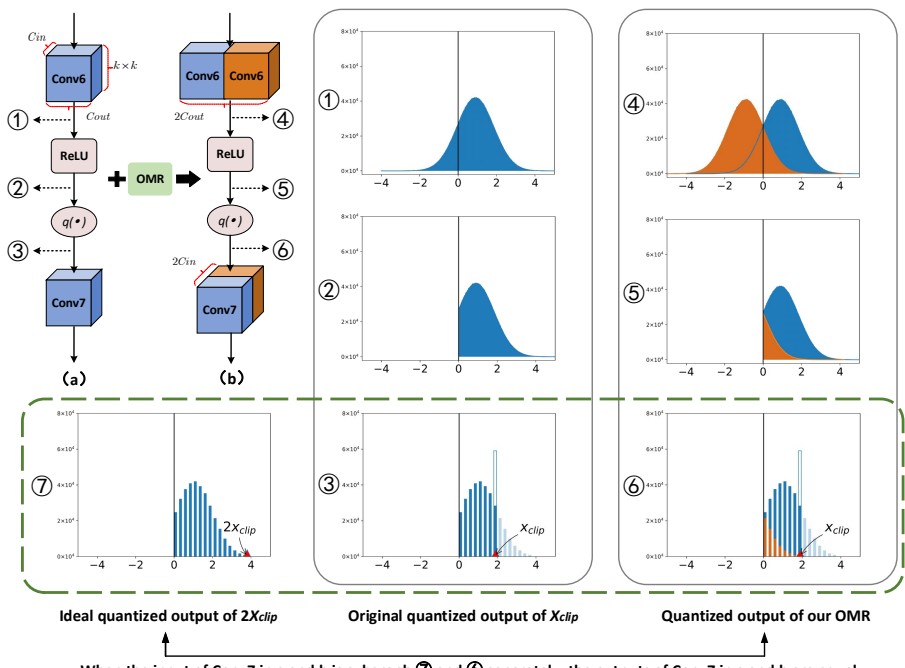

Figure 3: (a) is a typical [*Conv+ReLU+Conv*] structure used in MobileNet-v2. $q(\cdot)$ denotes quantization. Intermediate features of (a) are shown in subgraphs ①②③, whose distributions are shown in the middle. With OMR equipped, structure (a) can be transformed to (b), whose intermediate feature distributions are drawn in ④⑤⑥. The orange channels are copied from the blue ones. When the input of Conv7 in a and b is subgraph ⑦ and ⑥ separately, the outputs of Conv7 in a and b are equal. The detailed proof can be seen in the supplementary. Thus N-bit activation in OMR-solvable structures can be expanded to (N+1)-bit equally.

where $c_{out}$ is the original number of output channels for activation $\boldsymbol{y}^l$. To achieve this operation, we can transform the original weight $\boldsymbol{W}^l$ and bias $\boldsymbol{b}^l$ to new ones, i.e. $\boldsymbol{W}^{l\prime}$ and $\boldsymbol{b}^{l\prime}$ as denoted by:

$$
\boldsymbol{W}_{i,j}^{l\prime} = \begin{cases} \boldsymbol{W}_{i,j}^l & \text{if} \quad 0 \le j < c_{out} \\ \boldsymbol{W}_{i,j-c_{out}}^l & \text{if} \quad c_{out} \le j < 2c_{out} \end{cases}, \boldsymbol{b}_j^{l\prime} = \begin{cases} \boldsymbol{b}_j^l & \text{if} \quad 0 \le j < c_{out} \\ \boldsymbol{b}_{j-c_{out}}^l - x_{clip} & \text{if} \quad c_{out} \le j < 2c_{out} \end{cases}
\tag{12}
$$

With the modified weight $\boldsymbol{W}^{l\prime}$ and bias $\boldsymbol{b}'$, we can get the results of the formula equation 11 by $\boldsymbol{y}^{l\prime} = \boldsymbol{W}^{l\prime}\boldsymbol{x}^l + \boldsymbol{b}^{l\prime}$, To ensure the final output is intact as original except for more outliers saved, $\boldsymbol{W}^{l+1}$ needs to be transformed as $\boldsymbol{W}^{l+1\prime}$:

$$
\boldsymbol{W}_{i,j}^{l+1\prime} = \begin{cases} \boldsymbol{W}_{i,j}^{l+1} & \text{if} \quad 0 \le i < c_{out} \\ \boldsymbol{W}_{i-c_{out},j}^{l+1} & \text{if} \quad c_{out} \le i < 2c_{out} \end{cases}
\tag{13}
$$

Such modifications on weights make an almost functionally identical network except that more outlier activation in range $(X_{clip}, 2X_{clip}]$ is preserved. Therefore, our OMR enlarges the quantization level and keep the same fine-grained quant-step under the same quantization bit. This is very beneficial to some lightweight models and low-bit settings.

Besides, not all channels present the same importance. Therefore, if we can apply our OMR merely to the channels whose outliers are most sensitive to final task loss, a more balanced FLOPs V.s. accuracy trade off can be obtained. Here we assume the proportion of selected channels in each layer as $k \in [0, 1]$, thus the number of selected channels is $c_{out} \cdot k$. With a given channel sensitivity criterion, all channels can be sorted in order and the top k percentage of sensitive channels are selected to apply our OMR. For simplification, we directly adopt the sum of the activation in range $[X_{clip}, 2X_{clip}]$ as each channel's sensitivity criterion. Detailed exploration can be seen in section 4. The overall optimization process of our DOMR is shown in Algorithm 1.

# 4 EXPERIMENTS

We evaluate our DOMR's efficiency on ImageNet (Deng et al., 2009) classification task and MS COCO (Lin et al., 2014) object detection task across various networks and bitwidth settings. All experiments are implemented using Pytorch (Paszke et al., 2019) in one NVIDIA Tesla V100. The calibration set consists of 1024 (256) unlabeled images randomly selected from the training set of ImageNet (MS COCO). For fair comparison, we adopt Adam optimizer and the same learning rate as as (Wei et al., 2022; Ma et al., 2023). 20k iterations is consumed for reconstruction process. By convention, the first and last layer are both quantized into 8 bits. Per-channel quantization is adopted for weight.

## 4.1 CLASSIFICATION ON IMAGENET AND OBJECT DETECTION ON MS COCO

Table 2: Acc@1 on ImageNet among current PTQ methods on various networks.

| Methods | W/A | Mobile-v2 | Res-18 | Reg-600 | Mnas2.0 |
|---|---|---|---|---|---|
| Full Prec. | 32/32 | 72.49 | 71.08 | 73.71 | 76.68 |
| LAPQ(Nahshan et al., 2021) | 4/4 | 49.70 | 60.30 | 57.71 | 65.32 |
| AdaRound(Nagel et al., 2020b) | 4/4 | 64.33 | 69.36 | - | - |
| AdaQuant(Hubara et al., 2021) | 4/4 | 47.16 | 69.60 | - | - |
| BRECQ(Li et al., 2021b) | 4/4 | 66.57 | 69.60 | 68.33 | 73.56 |
| QDROP(Wei et al., 2022) | 4/4 | 68.84 | 69.62 | 71.18 | 73.71 |
| PD-Quant (Liu et al., 2023) | 4/4 | 68.33 | 69.30 | 71.04 | 73.30 |
| MRECG (Ma et al., 2023) | 4/4 | 68.84 | 69.46 | 71.22 | - |
| NWQ (Wang et al., 2022) | 4/4 | 69.14 | 69.85 | 71.92 | 74.60 |
| **DJOS(ours)** | 4/4 | 69.22 | 69.90 | 71.98 | 74.82 |
| **DJOS+OMR$_{0.5}$(ours)** | 4/4 | **69.46**±0.12 | **70.11**±0.05 | **72.21**±0.09 | **74.93**±0.22 |
| BRECQ(Li et al., 2021b) | 3/3 | 23.41 | 65.87 | 55.16 | 49.78 |
| QDROP(Wei et al., 2022) | 3/3 | 57.98 | 66.75 | 65.54 | 66.81 |
| PD-Quant (Liu et al., 2023) | 3/3 | 57.64 | 66.12 | 65.09 | 64.88 |
| MRECG (Ma et al., 2023) | 3/3 | 58.40 | 66.30 | 66.08 | - |
| NWQ (Wang et al., 2022) | 3/3 | 61.24 | 67.58 | 67.38 | 68.85 |
| **DJOS(ours)** | 3/3 | 63.57 | 67.71 | 67.72 | 69.60 |
| **DJOS+OMR$_{0.5}$(ours)** | 3/3 | **64.63**±0.18 | **68.20**±0.07 | **68.83**±0.09 | **71.65**±0.21 |
| BRECQ(Li et al., 2021b) | 2/2 | 0.24 | 42.54 | 3.58 | 0.61 |
| QDROP(Wei et al., 2022) | 2/2 | 13.05 | 54.72 | 41.47 | 28.77 |
| PD-Quant (Liu et al., 2023) | 2/2 | 13.67 | 53.14 | 40.92 | 28.03 |
| MRECG (Ma et al., 2023) | 2/2 | 14.44 | 54.46 | 43.67 | - |
| NWQ (Wang et al., 2022) | 2/2 | 26.42 | 59.14 | 48.49 | 41.17 |
| **DJOS(ours)** | 2/2 | 31.43 | 60.09 | 51.32 | 45.08 |
| **DJOS+OMR$_{0.5}$(ours)** | 2/2 | **39.35**±1.76 | **61.58**±0.12 | **55.05**±0.08 | **51.44**±1.21 |

To show the generalizability of our method, various neural structures are experimented on ImageNet classification task, including ResNet-18, MobileNet-v2 (Sandler et al., 2018), RegNetX-600MF (Radosavovic et al., 2020) and MnasNet2.0 (Tan et al., 2019). For some current PTQ methods which do not report their quantization results in some low-bit settings, we re-implement them based on their open source codes with unified settings. The average experimental results over 5 runs are summarized in Table 2. The "**DJOS**" in the table indicates that only DJOS in our method is used. The "**DJOS+OMR$_{0.5}$**" indicates that both DJOS and OMR are adopted, where the proportion of selected important channels is set as $k = 0.5$ in OMR. Most of the existing methods have good performance results in W4A4 setting. It can be observed that our method provides an accuracy improvement about $0 \sim 1\%$ compared to the strong baseline methods including NWQ (Wang et al., 2022), MRECG (Ma et al., 2023). In W3A3, our method improve Mobile-v2 by 3.39%, Reg-600 by 1.45% and Mnas2.0 by 2.80%. In W2A2, BRECQ shows nearly 0% Acc@1 on Mobile-v2 and Mnas2.0. However, our method still far outperforms NWQ by more than 10% on Mobile-v2, and Mnas2.0.

For object detection, we choose two-stage Faster RCNN (Ren et al., 2015) and one-stage RetinaNet (Lin et al., 2017), where Res-18, Res-50 and Mobile-v2 are selected as backbones respec-

Table 3: mAP for object detection on MS COCO.

| Methods | W/A | Faster RCNN | | RetinaNet | |
|---|---|---|---|---|---|
| | | Res-50 | Res-18 | Res-50 | Mobile-v2 |
| FP32 | 32/32 | 40.26 | 34.91 | 37.39 | 33.31 |
| BRECQ (Li et al., 2021a) | 4/4 | 37.19 | 33.41 | 34.67 | 29.81 |
| QDROP (Wei et al., 2022) | 4/4 | 38.53 | 33.57 | 35.81 | 31.47 |
| NWQ (Wang et al., 2022) | 4/4 | 38.54 | 33.63 | 35.98 | 31.81 |
| **DJOS(ours)** | 4/4 | 38.60 | 33.83 | 36.01 | 31.89 |
| **DJOS+OMR$_{0.5}$ (ours)** | 4/4 | **38.93** | **34.02** | **36.05** | **32.11** |
| QDROP (Wei et al., 2022) | 3/3 | 33.49 | 31.21 | 32.13 | 27.55 |
| NWQ (Wang et al., 2022) | 3/3 | 35.25 | 31.88 | 32.45 | 28.43 |
| **DJOS(ours)** | 3/3 | 35.68 | 32.13 | 32.50 | 28.82 |
| **DJOS+OMR$_{0.5}$ (ours)** | 3/3 | **36.44** | **32.51** | **33.35** | **29.65** |
| QDROP (Wei et al., 2022) | 2/2 | 21.05 | 21.95 | 20.27 | 12.01 |
| NWQ (Wang et al., 2022) | 2/2 | 25.01 | 23.92 | 22.95 | 16.21 |
| **DJOS(ours)** | 2/2 | 25.07 | 26.15 | 24.15 | 17.93 |
| **DJOS+OMR$_{0.5}$ (ours)** | 2/2 | **29.73** | **27.26** | **26.29** | **20.11** |

Table 5: Acc@1 on ImageNet for ViTs.

| Methods | W/A | ViT-S | ViT-B | DeiT-S | DeiT-B |
|---|---|---|---|---|---|
| FP32 | 32/32 | 81.39 | 84.54 | 79.80 | 81.80 |
| PTQ4ViT (Yuan et al., 2022) | 4/4 | 42.57 | 30.69 | 34.08 | 64.39 |
| APQ-ViT (Ding et al., 2022) | 4/4 | 47.95 | 41.41 | 43.55 | 67.48 |
| NWQ (Wang et al., 2022) | 4/4 | 57.79 | 56.87 | 65.76 | 76.06 |
| **DJOS(ours)** | 4/4 | **58.09** | **57.21** | **66.23** | **76.18** |

Table 6: Acc@1 with same FLOPs on ImageNet

| Methods | W/A | Mobile-v2 | Res-18 | Reg-600 | Mnas2.0 |
|---|---|---|---|---|---|
| Ori | 32/32 | 72.49 | 71.08 | 73.71 | 76.68 |
| Channels-plus | 32/32 | 74.85 | 71.80 | 75.37 | 78.57 |
| **OMR$_{0.5}$ + Ori** | 4/2 | 48.62 | 63.70 | 62.73 | 58.45 |
| NWQ + Channels-plus | 4/2 | 43.24 | 62.09 | 60.20 | 53.23 |
| **OMR$_{0.5}$ + Ori** | 2/2 | 39.35 | 61.58 | 55.05 | 51.44 |
| NWQ + Channels-plus | 2/2 | 32.02 | 60.64 | 52.73 | 47.24 |

Table 7: Acc@1 among different $k$ of OMR

| Methods | W/A | Mobile-v2 | Res-18 | Reg-600 | Mnas2.0 |
|---|---|---|---|---|---|
| DJOS(ours) | 2/2 | 31.43 | 60.09 | 51.32 | 45.08 |
| DJOS+OMR$_{0.3}$(ours) | 2/2 | 36.33 | 61.25 | 53.79 | 49.35 |
| DJOS+OMR$_{0.5}$(ours) | 2/2 | 39.35 | 61.58 | 55.05 | 51.44 |
| DJOS+OMR$_{0.7}$(ours) | 2/2 | 39.75 | 61.96 | 55.91 | 53.35 |
| **DJOS+OMR$_{1.0}$(ours)** | 2/2 | **41.65** | **62.11** | **57.24** | **54.40** |

Table 4: Acc@1 on ImageNet of OMR and OCS.

| Network | Methods | W/A | Acc | W/A | Acc | Weight size |
|---|---|---|---|---|---|---|
| Res-50 | OCS$_{0.1}$ | 4/4 | 6.7 | 2/2 | 0.1 | 1.1x |
| Res-50 | **OMR$_{0.1}$** | 4/4 | **75.50** | 2/2 | **59.68** | 1.1x |

tively. As (Wei et al., 2022; Li et al., 2021b), we quantize the input and output layers of the network to 8 bits and do not quantize the head of the detection model, but the neck (FPN) is quantized. The experimental results are shown in Table 3. In W3A3 setting, our method improves the mAP of Res-50-based Faster RCNN by 1.19% and Mobile-v2-based RetinaNet by 1.22%. In harder W2A2 setting, our method achieves more than 3% improvement over the current best method across all four experimental networks, which obtains a 4.72% improvement on Res-50-based Faster RCNN.

## 4.2 OMR V.S. CHANNELS-INCREASED NWQ UNDER THE SAME FLOPs.

For OMR$_{0.5}$, it brings extra 50% channels only for OMR-sovlable structures. Thus we add the same number of channels on the same layers in FP32 networks. Then we train these new FP32 networks from scratch with timm (Wightman, 2019) training pipeline. As the second row of Table 6, extra channels bring about 2.4%, 0.8%, 1.7% and 1.9% gain for MobileNet-v2, ResNet-18, RegNetX-600MF and MnasNet2.0 FP32 networks. Since most PTQs, like NWQ, do not change FLOPs of networks, it is fair to compare OMR$_{0.5}$ on original FP32 networks to NWQ on new FP32 networks under the same FLOPs. In W4A2 and W2A2 settings, OMR$_{0.5}$ achieves better performance across all these networks, especially for MoblieNet-V2, about 9.1% better in W4A2 and 8.8% in W2A2 setting. **Therefore, if we want to improve FP32 performance by adding more channels on networks, the gain will also get lost due to outlier clipping in most PTQ methods.** Differently, with these important outliers saved, our OMR achieves better performance.

## 4.3 COMPARISON BETWEEN OMR AND OCS UNDER THE SAME FLOPs

OCS (Zhao et al., 2019) also proposed to duplicate channels containing outliers. However, as description from themselves, OCS failed to quantize activation into low bits. Different from OCS's halving operation, our OMR's migration do not squeeze all values into a narrower range. Thus OMR achieves much better performance for activation quantization. Their practical extra-introduced cost is the same. The fair comparison with expanding ratio 0.1 is shown in Table 4.

**Algorithm 1:** PTQ using DOMR optimization

**Input:** Pretrained FP32 Model $\{W^l\}_{l=1}^N$; calib input $x^l$; the ratio of the selected channel $k$.
**Params:** x's quant-step $s_x$; W's quant / dequant-step, adaround param: $s_w, s'_w, \alpha$.

- - - - - - - - - - - - - - - - - - - - - - - - - -

$1_{st}$ Stage: Iterative MSE Optimization for $s_w$, then decouple $s_w$ and $s'_w$, and freeze $s_w$.

- - - - - - - - - - - - - - - - - - - - - - - - - -

$2_{nd}$ Stage: Jointly Optimize $s'_w, s_x, \alpha$, then OMR
**for** $j = 1$ to $T$-iteration **do**

    # Modify weight by OMR with equation 12
    $W^{i\prime} \leftarrow concat(W^i, W^i)$;
    $b^{i\prime} \leftarrow concat(b^i, b^i - x_{clip})$
    # Modify next layer $W^{i+1}$ as equation 13
    $W^{i+1\prime} \leftarrow concat(W^{i+1}, W^{i+1})$;
    # Forward to get FP32 and quantized output
    $\hat{W}^{i\prime} \xleftarrow{s_x,s_w,\alpha,s'_w} W^{i\prime}, \hat{x}^i \xleftarrow{s_x} x^i$ as equa4, 2
    $x^{i+1\prime} = W^{i\prime}x^i + b^{i\prime}$; $\hat{x}^{i+1\prime} = \hat{W}^{i\prime}\hat{x}^i + b^{i\prime}$
    $\Delta = x^{i+1\prime} - \hat{x}^{i+1\prime}$
    # Optimize $s_x, s'_w, \alpha$ as equation 6 with $\Delta$

- - - - - - - - - - - - - - - - - - - - - - - - - -

**Output:** Quantized model

Table 8: Effect Visualization of DJOS

| iters | 0 | 5k | 10k | 15k | 20K |
|---|---|---|---|---|---|
| $S_{w1}$ | 0.544 | 0.544 | 0.544 | 0.544 | 0.544 |
| $S_{w1}\prime$ | 0.544 | 0.508 | 0.460 | 0.444 | 0.442 |
| $S_{w2}$ | 0.943 | 0.943 | 0.943 | 0.943 | 0.943 |
| $S_{w2}\prime$ | 0.943 | 0.902 | 0.790 | 0.796 | 0.795 |
| Loss: a single $S_w$ | 107 | 59.3 | 48.2 | 55.2 | 50.7 |
| Loss: decoupled $S_w, S_w\prime$ | 107 | **54.4** | **43.2** | **51.1** | **46.5** |

Table 9: Gain of DJOS Varies on Different Networks and bitwidth settings

| Methods | W/A | Mobile-v2 | Res-18 | Reg-600 | Mnas2.0 |
|---|---|---|---|---|---|
| DJOS | 2/2 | **5.01**↑ | 0.95↑ | 2.93↑ | 3.91↑ |
| DJOS | 3/3 | **2.33**↑ | 0.13↑ | 0.34↑ | 0.75↑ |

Table 10: OMR V.s. Actual Bit+1

| Methods | W/A | Mobile-v2 | Res-18 | Reg-600 |
|---|---|---|---|---|
| DJOS | 4/2 | 41.77 | 61.77 | 59.56 |
| DJOS+OMR$_{0.5}$ | 4/2 | 48.62 | 63.70 | 62.73 |
| DJOS+OMR$_{1.0}$ | 4/2 | 52.08 | 64.74 | 65.16 |
| DJOS$_{bit+1}$ | 4/**3** | 52.08 | 64.74 | 65.16 |

Table 11: Overall Cost Comparison with Existing PTQs.

| Criterion | Importance to Inference | MRECG | NWQ | DJOS | DJOS+OMR |
|---|---|---|---|---|---|
| Extra Params/FLOPs/Mem in Fake Quant | Low | ✓ | ✓ | ✓ | ✓ |
| Quant Time in Fake Quant | High | < 1GPU-Hours | < 1GPU-Hours | < 1GPU-Hours | < 1GPU-Hours |
| Extra Params/FLOPs/Mem in Real Quant | High | ✗ | ✗ | ✗ | ✓ |

## 4.4 Ablation Study on ImageNet

1) Explore different proportion of selected $k$ in OMR. Table 7 with $k$ in [0,0.3,0.5,0.7,1.0] shows that performance improves with OMR applied, and performance gain improves more as $k$ increases.

2) Visualize the effect of DJOS on quant/dequant-step and its different gain on different networks and bit settings. As Table 8, DJOS with decoupled quant-step converges to a lower loss than original single quant-step. As Table 9, DJOS gains more on lighter-weight models and lower bitwidth.

3) Compare OMR with actual bit+1 on OMR-solvable structures for activation quantization. As Table 10, they perform the same.

4) The overall cost comparison with existing PTQs is shown on Table 11.

## 5 Conclusion

In this paper, we propose a novel PTQ approach called DOMR, including DJOS and OMR. DJOS propose to decouple the single quant-step into a quant-step and a dequant-step, then jointly optimize dequant-step only. OMR migrates the clipped outlier activation that is non-trivial into safe clipping range based on the nature of CNN structure and PTQ's clipping operation. Thus $b$-bit activation in OMR-solvable structures can be expanded to $(b + 1)$ bits equally, which equals to breaking the hardware's bitwidth shackle. Although OMR brings extra computation, it performs much better compared to other PTQs under the same FLOPs. Thus OMR is especially helpful on a predefined bitwidth accelerator. Experiments demonstrate DOMR establishes a new SOTA for PTQ.

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
