# DECOUPLE QUANTIZATION STEP AND OUTLIER-MIGRATED RECONSTRUCTION FOR PTQ

## 1 PROOF OF OMR: $b$-BIT WITH OMR EQUALS TO $b+1$-BIT

As shown in Figure 3 of the paper, ⑦ is ideal quantized output with $2X_{clip}$. ⑥ is the quantized outputs with OMR, whose clipping range is $X_{clip}$. When the input of conv7 in (a) and (b) is ⑦ and ⑥, the output of Conv7 in a and b are equal. The proof is shown as follows. $i$ is input channel index. $l$ denotes $l$-th layer.

The output of Conv7 in (a) can be denoted as:

$$y^{l+1} = \sum_{i=0}^{c_{out}} W_i^{l+1} clip(\hat{x}_i^{l+1}, 0, 2X_{clip})$$

The input of Conv7 in (b), subgraph 6, can be denoted as:

$$\hat{x}_i^{l+1\prime} = \begin{cases} \hat{x}_i^{l+1} & \text{if} \quad 0 \le i < c_{out} \\ \\ \hat{x}_{i-c_{out}}^{l+1} - X_{clip} & \text{if} \quad c_{out} \le i < 2c_{out} \end{cases}$$

The weight of Conv7 in (b) can be denoted as:

$$W_i^{l+1\prime} = \begin{cases} W_i^{l+1} & \text{if} \quad 0 \le i < c_{out} \\ \\ W_{i-c_{out}}^{l+1} & \text{if} \quad c_{out} \le i < 2c_{out} \end{cases}$$

Thus the output of Conv7 in (b) can be denoted as:

$$
\begin{aligned}
y^{l+1}\prime &= W_i^{l+1\prime} * \hat{x}_i^{l+1\prime} \\
&= \sum_{i=0}^{c_{out}} W_i^{l+1} clip(\hat{x}_i^{l+1}, 0, X_{clip}) + \sum_{i=c_{out}}^{2c_{out}} w_{i-c_{out}}^{l+1} * clip(\hat{x}_{i-c_{out}}^{l+1} - X_{clip}, 0, X_{clip}) \\
&= \sum_{i=0}^{c_{out}} W_i^{l+1} clip(\hat{x}_i^{l+1}, 0, X_{clip}) + \sum_{i=0}^{c_{out}} W_i^{l+1} clip(\hat{x}_i^{l+1} - X_{clip}, 0, X_{clip}) \\
&= \sum_{i=0}^{c_{out}} W_i^{l+1} [clip(\hat{x}_i^{l+1}, 0, X_{clip}) + clip(\hat{x}_i^{l+1} - X_{clip}, 0, X_{clip}) + X_{clip} - X_{clip}] \\
&= \sum_{i=0}^{c_{out}} W_i^{l+1} [clip(\hat{x}_i^{l+1}, 0, X_{clip}) + clip(X_i^{l+1}, X_{clip}, 2X_{clip}) - X_{clip}] \\
&= \sum_{i=0}^{c_{out}} W_i^{l+1} [clip(\hat{x}_i^{l+1} + X_{clip}, X_{clip}, 3X_{clip}) - X_{clip}] \\
&= \sum_{i=0}^{c_{out}} W_i^{l+1} clip(\hat{x}_i^{l+1}, 0, 2X_{clip}) = y^{l+1}
\end{aligned}
$$

## 2 COMPARISON VISUALIZATION OF OCS AND OMR

As shown in Fig1(notion the value changes in x-axis and the number of quantization levels), OCS squeezes outliers into a narrower range, which also squeezes inner values into a narrow. Thus OCS saves outliers at the sacrifice of the precision of inner values. However, our OMR saves outliers while preserves the precision of inner value as before. OMR indeed enlarges quantization levels and equals to earning one more unavailable bit, which is extremely helpful in 2,3-bit quantization.

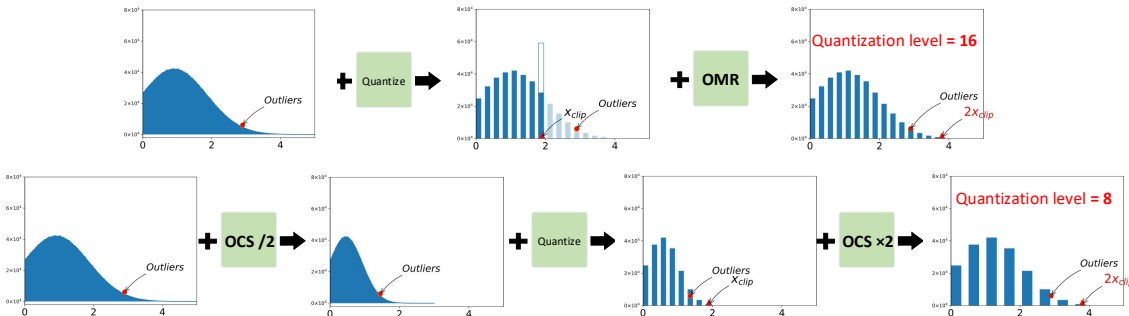

Figure 1: Difference between OCS and OMR

## 3 DETAILED ALGORITHM OF OUR DOMR→DJOS+OMR

---

**Algorithm 1:** PTQ using DOMR optimization

---

**Input:** Pretrained FP32 Model $\{W^l\}_{i=1}^N$; a batch of calibration input $x$; optim iteration $T$.
**Params:** quant-step of activation $s_x$; quant-step, dequant-step and AdaRound parameter of
weight $s_w$, $s'_w$ and $\alpha$.

- - - - - - - - - - - - - - - - - - - - - - - - - - - -

The $1_{st}$ Stage: $s_w$ initialization + $s_w$ decoupling.
       1.1 Iterative MSE Minimization (single $s_w$) as equation1 to find a suitable $s_w$
       1.2 Decouple $s_w$ into $s_w$ and $s'_w$, with $s_w$ fixed and $s'_w$ learnable

- - - - - - - - - - - - - - - - - - - - - - - - - - - -

The $2_{nd}$ Stage: OMR+DJOS. Outlier-Migrated Reconstruction(OMR) with Joint Optimization
       for $s'_w$, $s_x$, AdaRound $\alpha$ (DJOS)
**for** $j = 1$ to $T$-iteration **do**
    **for** each layer $i = 1$ to $N$ **do**
        # Modify weight $W^i$ to $W^{i\prime}$ for outlier feature migration as equation 2
        $W^{i\prime} \leftarrow concat(W^i, W^i); b^{i\prime} \leftarrow concat(b^i, b^i - x_{clip})$
        # Modify next layer $W^{i+1}$ for outlier feature migration as equation 3
        $W^{i+1\prime} \leftarrow concat(W^{i+1}, W^{i+1})$;
        # Fake quantize $W^i$ and $x^i$ as equation 4, equation 5;
        $\hat{W}^{i\prime} \xleftarrow{s_w, \alpha, s'_w} W^{i\prime}, \hat{x}^i \xleftarrow{s_x} x^i$
        # Forward to get FP32 and quantized output
        $x^{i+1\prime} = W^{i\prime}x^i + b^{i\prime}$;
        $\hat{x}^{i+1\prime} = \hat{W}^{i\prime}\hat{x}^i + b^{i\prime}$;
        # Calculate quantization error $\Delta_i$
        $\Delta_i = ||x^{i+1\prime} - \hat{x}^{i+1\prime}||_F^2$
        # Iterate the next layer.
    # Sum all layer's quantization error $\Delta_i$ for optimization.
    $\Delta = \sum \Delta_i$
    # Joint Optimization with $\Delta$ to update $s_x$, $s'_w$ and $\alpha$ as equation 6.

- - - - - - - - - - - - - - - - - - - - - - - - - - - -

**Output:** Quantized model

---

$$\hat{\boldsymbol{w}} = clip(\lfloor \frac{\boldsymbol{w}}{s_w} \rceil; w_l, w_u) \cdot s_w, \quad \min_{\hat{\boldsymbol{w}}} ||\hat{\boldsymbol{w}} - \boldsymbol{w}||_F^2 \tag{1}$$

$$\boldsymbol{W}_{i,j}^{l\prime} = \begin{cases} \boldsymbol{W}_{i,j}^l & \text{if} \quad 0 \le j < c_{out} \\ \boldsymbol{W}_{i,j-c_{out}}^l & \text{if} \quad c_{out} \le j < 2c_{out} \end{cases}, \boldsymbol{b}_j^{l\prime} = \begin{cases} \boldsymbol{b}_j^l & \text{if} \quad 0 \le j < c_{out} \\ \boldsymbol{b}_{j-c_{out}}^l - x_{clip} & \text{if} \quad c_{out} \le j < 2c_{out} \end{cases} \tag{2}$$

$$\boldsymbol{W}_{i,j}^{l+1\prime} = \begin{cases} \boldsymbol{W}_{i,j}^{l+1} & \text{if} \quad 0 \le i < c_{out} \\ \boldsymbol{W}_{i-c_{out},j}^{l+1} & \text{if} \quad c_{out} \le i < 2c_{out} \end{cases} \tag{3}$$

$$\hat{\boldsymbol{w}} = clip(\lfloor \frac{\boldsymbol{w}}{s_w} \rfloor + h(\boldsymbol{\alpha}); w_l, w_u) \cdot s'_w \tag{4}$$

$$\hat{\boldsymbol{x}} = clip(\lfloor \frac{\boldsymbol{x}}{s_x} \rceil; x_l, x_u) \cdot s_x \tag{5}$$

$$\min_{s'_w, \boldsymbol{\alpha}, s_x} ||\hat{\boldsymbol{W}}\hat{\boldsymbol{x}} - \boldsymbol{W}\boldsymbol{x}||_F^2 \tag{6}$$

## 4   OMR ON GELU, H-SWISH

Except for Conv-ReLU-Conv and positive nonlinear function like ReLU6/Sigmoid, the core of OMR, migrating outliers into safe clipping range then compensating in the following layers, can be extented to other structures like Conv-Conv, Linear-Linear and nonlinear function like h-swish, GeLU. The migration comparison is visualized as Figure 2.

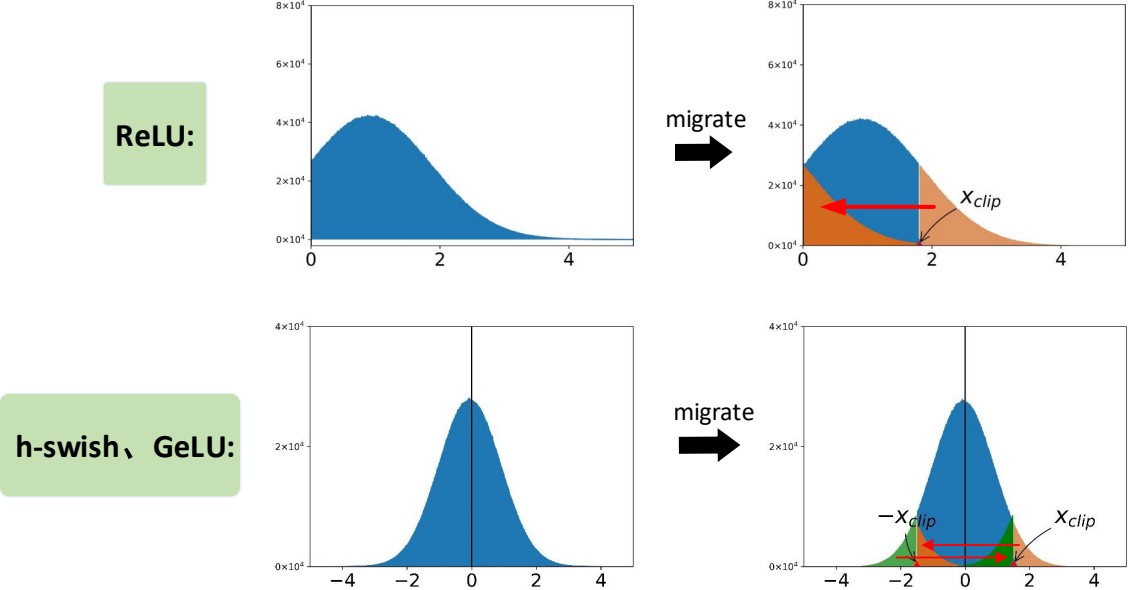

Figure 2: OMR on ReLU and GeLU/h-swish