# OpenReview forum: "DECOUPLE QUANTIZATION STEP AND OUTLIER-MIGRATED RECONSTRUCTION FOR PTQ"
_ICLR.cc/2024/Conference — Submitted to ICLR 2024_

### Official Review · Reviewer_wiuF · 2023-10-30

**Soundness:** 3 good
**Presentation:** 1 poor
**Contribution:** 3 good
**Rating:** 6
**Confidence:** 4

**Summary:**

PTQ (Post-Training Quantization) is a technique that exports a pre-trained full precision neural network into low-bit. It is a promising way to reduce size of neural networks and costs of inference. However, PTQ can cause accuracy degradation with extremely low-bit settings. To solve this problem, the paper proposes two methods that are for improving PTQ process.

In the quantization process, weight parameters of a target network are quantized before inference. The paper tackles that there isn’t any previous work that focuses on how to quantize weight parameters better. With a simple yet clear experiment, the paper shows that decoupling quant step and dequant step, and fine-tuning dequant step only lead to performance gain. In addition, the paper points out that outlier activation values that are clipped out after quantization affect performance a lot. To utilize these outlier values, the paper proposes OMR (Outlier-Migrated Reconstruction) that adds several channels to filters and operates them with outlier values which are shifted into safe clipping ranges. It has the same effect as increasing the number of bits.

The paper surpasses other previous works only with the decoupling quant step and dequant step. In addition, with OMR, the paper shows that higher accuracy can be obtained by sacrificing efficiency.

**Strengths:**

- The paper explains the proposed method well with several figures and formulas.
- The paper shows performance improvement with simple methods that are easy to follow.
- The paper presents various experiments for presenting the efficiency of the proposed method.

**Weaknesses:**

- OMR leads to enlarging the size of neural networks. Therefore, the application of the proposed method can be limited. However, the paper doesn’t provide analysis in terms of costs which helps in understanding the trade-off between the size of a model and its performance.
- It doesn’t seem appropriate that the arrangements and citations of figures and tables. For instance, the first citation of Figure 2 is on page 2, and Figure 2 is on page 5.

**Questions:**

- In Outlier-Migrated Reconstruction, how can the sensitivity of each channel be measured, if only a portion of channels will be added?
- It seems that OMR is designed to mitigate outlier of activations and outlier of weight parameters can’t be utilized with this method. Does OMR help mitigate outlier of weight parameters?

---

> ### Author Response · Authors · 2023-11-22
> **OMR’s Computational Cost and Sensitivity Criterion,  re-polished Figure and OMR on Weight**
>
> Thanks for your instructive review.
>
> ## 3.1 OMR’s Computational Cost
>
> Our OMR does no need the Fake Quant process. It can be finished in **1 CPU-second** based on offline methematical transformation. It introduces extra FLOPs during Real Quant.
>
> - For k=0.2 or k=0.5, where we save 20% or 50% channles' outliers, if
> FLOPs of original OMR-Solvalble structures is 10M, then FLOPs of
> OMR-applied ones is 12M or 15M.
> - Trade-off analysis on MobileNet-V2 between FLOPs/Params and its performance is as follows. We denote the FLOPs 327M, params 3.5M of MobileNet-V2 on W2A2 as 1.0, on which of other methods is based. Note that "W2A3" can not run on 2-bit hardware while our **OMR can run on 2-bit hardware with W2A3 performance** as said on our paper PDF.
>
>     | MobileNet-V2 | W2A2 | FLOPs | Params | Run on 2bit Hardware |
>     | --- | --- | --- | --- | --- |
>     | NWQ+Ori_Net | 26.42 | 1.0 | 1.0 | √ |
>     | NWQ+Channel-plus -Net | 32.02 | 2.0 | 2.0 | √ |
>     |**OMR_0.0+Ori-Net(DJOS)**|**31.43**|**1.0**|**1.0**|√|
>     | OMR_0.3+Ori-Net** | 36.33 | 1.3 | 1.3 | √ |
>     | **OMR_0.5+Ori-Net** | **39.35** | 1.5  | 1.5 | √ |
>     | OMR_0.7+Ori-Net | 39.75 | 1.7 | 1.7 | √ |
>     | OMR_1.0_Ori-Net | 41.65 | 2.0 | 2.0 | √ |
> - **To balance FLOPs-vs-Acc trade-off, OMR can be only applied
> on the most important channels of the most important OMR-Solvable
> structures with a given FLOPs limit,** which can be explored in the future.
>
> ## 3.2 Arrangements and Citations of Figures and Tables
>
> Due ICLR’s limited space, the placement and citation of figures and tables is a little far. To make full use of limited PDF space, we place experiment tables as together as possible. It is a better choice to place Figure 2 on page 2, thus readers can better understand our OMR’s motivation. We has re-arranged our Figures as suggested, which can be seen on our re-polished paper PDF.
>
> ## 3.3 Sensitivity Criterion on OMR
>
> If we only select the most sensitive channels on the most sensitive OMR-Solvable structures, OMR can be more efficient.
>
> The sensitivity criterion on channels is as follows, where we choose the sum of each channels’ activation in outlier range$[X_{clip}, 2X_{clip}]$, then we sort each channel in order and choose the top-k percent for OMR.
>
> $$
> S_i=\sum_j{||x_{ij}||^2},
> $$ where $x_{i,j}$ is the $j$-th values in range $(X_{clip}, 2X_{xlip}]$  of $i$-th channel.
> $$
> S^\*={sort}(S_i|i=1,...,N), S^{\*k}={Top\\_K}(S^{\*})
> $$
>
>
> Other sensitivity criterion can also be applied, which can be explored as a future work.
>
> ## 3.4 OMR can be Used on Weight
>
> It is right that OMR can also be used on weight like activation. **The core of OMR, migrating outliers into safe clipping range then compensating in the following layers, can be extended to weight**. Detailed analysis is as follows,
>
> - When OMR is used on activation, we want to save outlier activation in range $[X_{clip}, 2X_{clip})$. According to equal mathematical transformation, we can transform activation modification into weight modification. We can modify the $i$-th and $i+1$-th weight as follows, thus a functionally-identical network is caused without retraining except for more outlier activation being preserved.
>
> $$
> \begin{equation}W_{i,j}^{l\prime} = \begin{cases}W_{i,j}^{l}, & \text{if } 0<j≤c_{out}; \\\\W_{i,j-c_{out}}^{l}, & \text{if } c_{out}<j≤2c_{out}.\end{cases}
> b_{j}^{l\prime} = \begin{cases}b_{j}^{l}, & \text{if } 0<j≤c_{out}; \\\\b_{j-c_{out}}^{l}-X_{clip}, & \text{if } c_{out}<j≤2c_{out}.\end{cases}
> \end{equation}
> $$
>
> $$
> \begin{equation}W_{i,j}^{l+1\prime} = \begin{cases}W_{i,j}^{l}, & \text{if } 0<i≤c_{out}; \\\\W_{i-c_{out},j}^{l}, & \text{if } c_{out}<i≤2c_{out}.\end{cases}
> \end{equation}
> $$
>
> - When OMR is used on weight $W$, we want to save outlier weight in range $[W_{clip}, 2W_{clip)}$. We can directly transform $i$-th and $i+1$-th weight as follows,
>
>     $$
>     \begin{equation}W_{i,j}^{l\prime} = \begin{cases}W_{i,j}^{l}, & \text{if } 0<j≤c_{out}; \\\\W_{i,j-c_{out}}^{l}-W_{clip}, & \text{if } c_{out}<j≤2c_{out}.\end{cases}
>     b_{j}^{l\prime} = \begin{cases}b_{j}^{l}, & \text{if } 0<j≤c_{out}; \\\\b_{j-c_{out}}^{l}, & \text{if } c_{out}<j≤2c_{out}.\end{cases}
>     \end{equation}
>     $$
>
>     $$
>     \begin{equation}W_{i,j}^{l+1\prime} = \begin{cases}W_{i,j}^{l}, & \text{if } 0<i≤c_{out}; \\\\W_{i-c_{out},j}^{l}, & \text{if } c_{out}<i≤2c_{out}.\end{cases}
>     \end{equation}
>     $$

---

> ### Author Response · Authors · 2023-11-22
>
> Dear Reviewer wiuF,
>
> Thanks for your valuable reviews.
>
> If you have any problem, we are online for response by the last second of 22nd November. We are reaching out to see if our response adequately addresses your concerns. We greatly value your comments and are committed to refining our work. Any further feedback would be helpful in polishing our final revisions.
>
> Thank you for your time and expertise.
>
> Sincerely,
>
> Authors

---

> ### Comment · Reviewer_wiuF · 2023-11-22
>
> Thank the authors for the answers to the reviewer's concerns and questions.
>
> Lastly, the reviewer will ask the authors one short question.
>
> I agree with the reviewer pf4t that OMR seems similar with OCS.
>
> What is the critical difference between OMR and OCS?

---

> ### Author Response · Authors · 2023-11-22
> **OMR V.s. OCS**
>
> Dear reviewer wiuF,
>
> Thanks for your valuable feedback.
>
> ## 1. OMR V.s. OCS
> Our OMR is motivated from OCS to overcome OCS's shortcoming of sacrificing the precision of inner values.
>
> It is true that when integer activation is  [2,4,8,16], OMR share the same precision as OCS as reviewer pf4t.
>
> However, when input is [2,4,8,15], OCS will split split [2,4,8,15]->[1,2,4,7]+[1,2,4,7] or [2,4,8,15]->[1,2,4,8]+[1,2,4,8], thus there will be **an rounding error** for 15->7\*2=14 or 15->8\*2=16, while OMR will split [2,4,8,15]->[1,2,4,8]+[1,2,4,7], thus no error is caused for 15->8+7.
>
> The main problem is that when the bin's number of inputs is larger than quantization levels, OCS saves outliers by sacrificing the precision of inner values.
>
> For a more detailed example with the whole fake quantization: x=[0.0, 0.1, 0.2, 0.3, *0.4*, *0.5*, *0.6*, *0.7*], step=0.1, 2-bit, Thus outliers is [0.4,0.5,0.6,0.7].
> - Normal 2-bit:
>   - $x_{int}=clip[round(x/step), 0, 3]=[0, 1, 2, 3, 3, 3, 3, 3]$ quantization levels: 4
>   - $\hat{x}=x_{int}\*step=[0.0, 0.1, 0.2, 0.3, 0.3, 0.3, 0.3, 0.3]$ quantization levels: 4
> - OCS 2-bit:
>   - $x_{int}=clip[round((x/2)/step), 0, 3]=[0, 0, 1, 1, 2, 2, 3, 3]$ quantization levels: 4
>   - $\hat{x}=(x_{int}\*step)*2=[0.0, 0.0, 0.2,0.2,  0.4, 0.4, 0.6,0.6]$ quantization levels: 4
> - OMR 2-bit:
>   - $x_{int}=concat\\{clip[round((x)/step), 0, 3], clip[round((x-0.4)/step), 0, 3]\\}=[[0, 1,  2, 3],[0,1,2,3]]$
>   - $\hat{x}=concat\\{x_{int}[0]\*step, x_{int}[1]*step+0.4]\\}=concat\\{[0.0, 0.1, 0.2, 0.3], [0.4,0.5, 0.6, 0.7]\\}=[0.0, 0.1, 0.2, 0.3, 0.4, 0.5, 0.6, 0.7]$ **quantization levels: 8**
>
> Therefore:
> - **OMR is closer to (here equal to) original fp32 values during fake-quant with larger quantization levels**.
>   - [0.0, 0.1, 0.2, 0.3, 0.4, 0.5, 0.6, 0.7] $\rightarrow$ [0.0, 0.1, 0.2, 0.3, 0.4, 0.5, 0.6, 0.7]
> - OCS sacrifices [0.0, 0.1, 0.2, 0.3, 0.4, 0.5, 0.6, 0.7] $\rightarrow$  [0.0, 0.0, 0.2,0.2,  0.4, 0.4, 0.6,0.6] during fake-quant.
>
> In conclusion:
> - OCS does not enlarge quantization levels. OCS saves outliers while sacrifices the precision of inner values.
> - Our OMR, for the first time, proposes to save more outliers while preserve the precision of inner values.
>
> **Comparison visualization is re-polished in supplementary PDF Sec2.**
>
> If you have any problem, we are online for response by the last second of 22nd November. We are reaching out to see if our response adequately addresses your concerns. We greatly value your comments and are committed to refining our work.
>
> Sincerely,
>
> authors

---

> ### Comment · Reviewer_wiuF · 2023-11-22
>
> Thanks for the answer to my last question.
> It helps understand OMR shows better expressibility compared to OCS.
> Therefore, I decided to raise my score to 6.

---

> > ### Author Response · Authors · 2023-11-23
> >
> > Dear reviewer wiuF,
> >
> > Thank you for your time and expertise.
> >
> > We are sincerely grateful for your recognition of the improvements we've implemented in our manuscript based on your feedback, and for your decision to raise the evaluation score.
> >
> > We greatly value your comments and are committed to refining our work.
> >
> > Sincerely,
> >
> > authors

---

### Official Review · Reviewer_pf4t · 2023-10-31

**Soundness:** 2 fair
**Presentation:** 3 good
**Contribution:** 1 poor
**Rating:** 3
**Confidence:** 5

**Summary:**

This paper studies the Post-training quantization (PTQ) of deep learning models. Two main methods are proposed for weight and activation PTQ. For weight PTQ, DJOS is proposed to use a fixed quantization step for quantization and a learned step size for de-quantization. For activation PTQ, Outlier-Migrated Reconstruction (OMR) is proposed to split a channel into multiple channels to solve the outlier activations. The proposed methods are evaluated on ImageNet task and COCO task across various networks and bitwidth settings.

**Strengths:**

1. The authors fully explore different settings of weight’s quant-step into five cases, and study the performance of different settings.
2. The Outlier-Migrated Reconstruction is straight-forward and effective for PTQ given a pre-defined bit-width.

**Weaknesses:**

1. My main concern is about the novelty. The proposed Outlier-Migrated Reconstruction is well studied by previous works [1].
2. It is hard for me to understand why DJOS works. If the quant-step is fixed and the dequant-step is learnable, it is the same with the learning process of BN. In other words, network quantization with BN has already a mechanism of learned dequant-step.
3. In the experiments, the baselines of different methods are not the same, making the comparison unfair.
4. It is better to study weight-only quantization (DJOS) and compare with previous methods to evaluate the effectiveness of DJOS.

[1] Improving Neural Network Quantization without Retraining using Outlier Channel Splitting, ICML, 2020.

**Questions:**

1. What's the performance of weight-only quantization using the proposed method?
2. Which pretrained models are used as baseline?
3. Could you provide the detailed quantization algorithm?

---

> ### Author Response · Authors · 2023-11-18
> **Novelty of OMR and DJOS, Detailed Algorithm**
>
> Thanks for your instructive review.
> ## 2.1 Novelty of OMR
> We compare OMR with OCS in Sec.4.3. It is true that OCS studied the outlier problem. However, the outlier problem is **not well studied by OCS** due to its shortcomings.
> - OCS does not enlarge quantization levels. OCS saves outliers **while sacrifices the precision of inner values**.
>     - OCS saves outliers by halving operation. OCS squeezes the outlier values into a narrower range, however, which also squeezes the inner values into a narrower range. Thus OCS save outliers at the sacrifice of the precision of inner values.
>     - OCS failed to help achieve better performance on low-bit quantization.
> - **Our OMR, for the first time, proposes to save more outliers while preserve the precision of inner values**. Thus OMR actually enlarge quantization levels and equals to earning one more unavailable bit, which is extremely helpful for 2,3-bit, quantization.
> - **Their comparison visualization can be seen in the supplementary PDF Sec-2**.
> ## 2.2 DJOS V.s. BN
> - Why DJOS works:
>     Previous methods use a single and fixed quant-step of weight $s_w$.
>     For the first time, we propose 1) to make $s_w$ learnable; 2) to decouple single $s_w$ into $s_w$ and $s^\prime_w$; 3) to make $s_w$ fixed and $s^\prime_w$ learnable.
>     - Why making $s_w$ learnable :
>         - In general, more params under necessity during training will benefit model's optimization[1], especially under a extremely low-bit param space. Experiments also demonstrate decoupling brings one more parameter and brings lower reconstruction loss (both training and evaluation).
>     - Why decoupling single $s_w$ into $s_w, s^\prime_w$ works:
>         - Since integer weight can be obtained before inference, we can safely decouple traditional single $s_w$ into $s_w, s^\prime_w$. Integer activation need real-time calculation, thus we can not decouple activation’s quant-step $s_x$.
>     - Why fixed $s_w$ and learnable $s^\prime_w$ works:
>         - PTQ is in shortage of enough labeled training set and long time optimization,  and STE’s mismatched gradient is a little far way from the floor operation. In extremely-low-bit quantization, these two problems make the optimization of $s_w$ extremely difficult. Thus a fixed $s_w$ is better than wrongly-updated $s_w$. Therefore, it will achieve better performance to make the dequant-step $s^\prime_w$ learnable only.
> - Comparison Between DJOS and BN:
>     - **It is not true to say “When we have BN, we do not need fake quantization”. They are two orthogonal methods, BN helps for fp32 model training and DJOS helps for low-bit quantization.**
>     - Correlation: From the point of math formula, DJOS share similar form as BN.
> $$
> \text{DJOS}:\hat{w}=s^\prime_w\cdot \\{\lfloor \frac{x}{s_w}\rfloor +h(\alpha)\\}\ \ \ \ \text{with $s_w$ fixed and $s^\prime_w$ learnable}
> $$
> $$BN: {y}=\gamma\cdot \frac{x-\mu}{\sigma}+\beta, \text{with $\mu,\sigma$ statistically updated and $\gamma, \beta$ learnable}
> $$
> - However, they are diffenrent in physical sense.
>     - BN is used to reduce the internal covariate shift, which **maps fp32 values(any distribution) into a $N(0,1)$ Gussian-like distribution then learn to recover them**. It is usually used on activation.
>     - DJOS is a fake quantization method, which **maps fp32 values into integer space then re-map them into concrete fp32 counterparts. DJOS does not change original distribution**. **DJOS is used on weight and can not be used on activation**, because integer weight can be obtained before inference while integer activation has to be computed on line during inference.
>
> [1] I. Goodfellow, 2016. Deep Learning. MIT Press
> ## 2.3 Baseline Pretrained Model
> We borrowed FP32 pretrained model from BRECQ(https://github.com/yhhhli/BRECQ),  also the same as QDROP and NWQ. Thus our comparison is fair.  PD-Quant states different FP32 Acc@1. Their FP32 Acc@1 on their paper  are as follows:
> | FP32 Acc@1| Mobile-V2 | Res-18 | Reg-600 | Mnas-2.0 |
> | --- | --- | --- | --- | --- |
> | DOMR(ours) | 72.49 | 71.08 | 73.71 | 76.68 |
> | BRECQ | 72.49 | 71.08 | 73.71 | 76.68 |
> | QDROP | 72.49 | 71.08 | 73.71 | 76.68 |
> | NWQ | 72.49 | 71.06 | 73.71 | 76.68 |
> | PD-Quant | 72.62 | 71.01 | 73.52 | 76.52 |
>
> We can see the FP32 Acc@1 for Mobile-V2, Res-18, Reg-600, Mnas-2.0 in PD-Quant is slightly different from BRECQ, QDROP, NWQ and ours. Their **FP32 difference is no more than 0.2 percentage**. However, our DOMR is **at least 1 percentage better than others, at least 2 percentage better than PD-Quant**,  in W3A3 and W2A2. Thus DOMR’s performance is still robust and it is fair to say DOMR set up a new SOTA for PTQ.
> ## 2.4 Weight-Only Quantization with DJOS
> We can see DJOS outperforms existing PTQ SOTA in weight-only quantization.
> |  | QDROP | PD-Quant | NWQ | DJOS |
> | --- | --- | --- | --- | --- |
> | W2A32-Res-18 | 66.02 | 66.66 | 67.11 | 67.83 |
> | W3A32-Res-18 | 70.04 | 70.13 | 70.16 | 70.38 |
> ## 2.5 Detailed Algorithm can be seen in the supplementary PDF Sec-3.

---

> > ### Comment · Reviewer_pf4t · 2023-11-22
> > **Comments to the author response**
> >
> > Thanks for the detailed response. But the response does not fully address my concerns.
> >
> > 1. My main concern is about the novelty of OMR compared with OCS. I understand there are some minor differences between OMR and OCS, however, these differences are not critical. The main idea is to split the channel into two parts, i.e., $WX=WX_1 + W’X_2$, however, how to split is not the key point. There are many other splitting ways. I also can’t agree with the authors about the precision sacrificing/preserving of inner values. The OCS is like to split [2,4,8,16]->[1,2,4,8]+[1,2,4,8], while OMR is like to split [2,4,8,16]->[2,4,8,8]+[0,0,0,8]. We can’t say OMR is more accurate than OCS for inner values.
> >
> > 2. I also understand the difference between fake quantization and BN. My concern is that the learnable dequantization-step plays the same role as the *gamma parameter* of BN, which is a learnable affine transformation of features.
> >
> > Based on the above, I would like to keep my rating.

---

> > > ### Author Response · Authors · 2023-11-23
> > >
> > > Dear reviewer pf4t,
> > >
> > > Thank you for your time and expertise.
> > >
> > > As we will not be able to provide additional results or modifications to our manuscript after November 22nd, we are reaching out to see if our response about OMR and OCS adequately addresses your concerns.
> > >
> > > If you have any problem about OMR, we are online for response by the last second of 22nd November.
> > >
> > > Further, our OMR's novelty is also approved just now by reviewer wiuF who has the same concern,
> > > >Official CommentReviewer wiuF23 Nov 2023, 07:24Everyone
> > >
> > > >Thanks for the answer to my last question. It helps understand **OMR shows better expressibility compared to OCS**. Therefore, I decided to raise my score to 6.
> > >
> > >  We greatly value your comments and are committed to refining our work.
> > >
> > > Sincerely,
> > >
> > > authors

---

> ### Author Response · Authors · 2023-11-22
> **Novelty of our OMR and DJOS**
>
> Dear Reviewer pf4t,
>
> Thanks for your valuable reviews.
>
> Except for comments we provide before, the novelty of our OMR and DJOS, and our contribution on both weight and activation quantization is as what Reviewer RfJv commented:
> >"The authors provide intriguing and valuable insights, particularly in the idea of enhancing both weights and activations quantization. The decoupling of weight step-sizes for optimization is a noteworthy contribution, and the notable performance enhancements achieved by introducing a single additional step-size in each layer is quite interesting."
>
> If you have any problem, we are online for response by the last second of 22nd November. We are reaching out to see if our response adequately addresses your concerns. Thank you for your time and expertise.
>
> Sincerely,
>
> Authors

---

> ### Author Response · Authors · 2023-11-22
> **OCS V.s. OMR, DJOS V.s. BN**
>
> Dear reviewer pf4t,
>
> Thanks for your valuable feedback.
>
> ## 1. OMR V.s. OCS
> Our OMR is motivated from OCS to overcome OCS's shortcoming of sacrificing the precision of inner values.
>
> It is true that when integer activation is  [2,4,8,16], OMR share the same precision as OCS.
>
> However, when input is [2,4,8,15], OCS will split split [2,4,8,15]->[1,2,4,7]+[1,2,4,7] or [2,4,8,15]->[1,2,4,8]+[1,2,4,8], thus there will be **an rounding error** for 15->7\*2=14 or 15->8\*2=16, while OMR will split [2,4,8,15]->[1,2,4,8]+[1,2,4,7], thus no error is caused for 15->8+7.
>
> The main problem is that when the bin's number of inputs is larger than quantization levels, OCS saves outliers by sacrificing the precision of inner values.
>
> For a more detailed example with the whole fake quantization: x=[0.0, 0.1, 0.2, 0.3, *0.4*, *0.5*, *0.6*, *0.7*], step=0.1, 2-bit, Thus outliers is [0.4,0.5,0.6,0.7].
> - Normal 2-bit:
>   - $x_{int}=clip[round(x/step), 0, 3]=[0, 1, 2, 3, 3, 3, 3, 3]$ quantization levels: 4
>   - $\hat{x}=x_{int}\*step=[0.0, 0.1, 0.2, 0.3, 0.3, 0.3, 0.3, 0.3]$ quantization levels: 4
> - OCS 2-bit:
>   - $x_{int}=clip[round((x/2)/step), 0, 3]=[0, 0, 1, 1, 2, 2, 3, 3]$ quantization levels: 4
>   - $\hat{x}=(x_{int}\*step)*2=[0.0, 0.0, 0.2,0.2, 0.4, 0.4, 0.6,0.6]$ quantization levels: 4
> - OMR 2-bit:
>   - $x_{int}=concat\\{clip[round((x)/step), 0, 3], clip[round((x-0.4)/step), 0, 3]\\}=[[0, 1,  2, 3],[0,1,2,3]]$
>   - $\hat{x}=concat\\{x_{int}[0]\*step, x_{int}[1]*step+0.4]\\}=concat\\{[0.0, 0.1, 0.2, 0.3], [0.4,0.5, 0.6, 0.7]\\}=[0.0, 0.1, 0.2, 0.3, 0.4, 0.5, 0.6, 0.7]$ **quantization levels: 8**
>
> Therefore:
> - **OMR is closer to (here equal to) original fp32 values during fake-quant with larger quantization levels**.
>   - [0.0, 0.1, 0.2, 0.3, 0.4, 0.5, 0.6, 0.7] $\rightarrow$ [0.0, 0.1, 0.2, 0.3, 0.4, 0.5, 0.6, 0.7]
> - OCS sacrifices [0.0, 0.1, 0.2, 0.3, 0.4, 0.5, 0.6, 0.7] $\rightarrow$  [0.0, 0.0, 0.2,0.2,  0.4, 0.4, 0.6,0.6] during fake-quant.
>
> In conclusion:
> - OCS does not enlarge quantization levels. OCS saves outliers while sacrifices the precision of inner values.
> - Our OMR, for the first time, proposes to save more outliers while preserve the precision of inner values.
>
> **Comparison visualization is re-polished in supplementary PDF Sec2.**
>
>
> ## 2. DJOS V.s. BN
> We can say learnable dequant-step plays a role of learnable affine transformation of features. However, during PTQ(NWQ, DDROP, AdaRound, Brecq), the BN has been merged into convolution to accelerate inference. Thus there has been no  learnable affine transformation parameters $\gamma$, thus our learnable dequant-step is not redundant. We need a learnable dequant-step to better transform integer activation back to concrete fp32 counterparts with less quantization error.
>
>
> If you have any problem, we are online for response by the last second of 22nd November. We are reaching out to see if our response adequately addresses your concerns. We greatly value your comments and are committed to refining our work.
>
> Sincerely,
>
> authors

---

> > ### Author Response · Authors · 2023-11-23
> > **OCS's original paper experiment Comparison**
> >
> > Dear reviewer pf4t,
> >
> > The table below is the cited Table 3 from **OCS's original paper[1]**. We can see **OCS performs even much worse than traditional MSE PTQ baseline**. When quantization bitwidth is from 8 to 4, the degradation of OCS is more obvious. The reason is that OCS saves outliers while sacrifices the precision of inner values, also sacrifices the precision of outlier values.
> >
> > >Table 3. ImageNet Top-1 validation accuracy with **activation quantization** – formatting is
> > identical to Table 2 except weight bits is kept at 8 while the activation bitwidth is changed. We
> > did not combine OCS with clipping due to ineffectiveness of OCS on activations.
> > | Network | Act  Bits |  | Clip |  |  | Clip  Best |  | OCS |  |
> > |:---:|:---:|:---:|:---:|:---:|:---:|:---:|:---:|:---:|:---:|
> > |  |  | None | MSE | ACIQ | KL |  | 0.01 | 0.02 | 0.05 |
> > | VGG16-BN (73.4) | 8 | 72.5 | **73.2** | 73.1 | 73.2 | 73.2 | 72.7 | 72.8 | 72.5 |
> > |  | 7 | 70.8 | **72.8** | **72.8** | 72.7 | 72.8 | 70.5 | 70.7 | 70.2 |
> > |  | 6 | 49.7 | 71.3 | **71.4** | 70.6 | 71.4 | 49.2 | 46.0 | 45.9 |
> > |  | 5 | 0.7 | **62.0** | 58.1 | 51.6 | 62.0 | 1.6 | 1.0 | 1.4 |
> > |  | 4 | 0.1 | **11.5** | 5.0 | 2.4 | 11.5 | 0.1 | 0.2 | 0.1 |
> > | ResNet-50 (76.1) | 8 | 75.5 | **75.9** | 75.9 | 75.8 | 75.9 | 75.6 | 75.5 | 75.7 |
> > |  | 7 | **75.4** | 75.3 | 75.2 | 75.3 | 75.4 | 74.1 | 74.5 | 74.1 |
> > |  | 6 | 62.6 | **73.5** | **73.5** | 72.8 | 73.5 | 63.3 | 63.3 | 63.6 |
> > |  | 5 | 5.7 | 63.7 | **65.4** | 56.7 | 65.4 | 10.0 | 12.6 | 6.0 |
> > |  | 4 | 0.1 | 9.0 | **20.6** | 7.2 | 20.6 | 0.1 | 0.1 | 0.1 |
> >
> >
> > [1] Improving Neural Network Quantization without Retraining using Outlier Channel Splitting, ICML, 2020.
> >
> > If you have any problem, we are online for response by the last second of 22nd November. We are reaching out to see if our response adequately addresses your concerns. We greatly value your comments and are committed to refining our work.
> >
> > Sincerely,
> >
> > authors

---

### Official Review · Reviewer_RfJv · 2023-10-31

**Soundness:** 3 good
**Presentation:** 3 good
**Contribution:** 2 fair
**Rating:** 6
**Confidence:** 4

**Summary:**

This paper presents a new method for improving the performance of Post-training Quantization (PTQ). The authors identify two key obstacles to overcoming the performance drop of PTQ in extremely low-bit settings: (i) Separate quantization step-size scale factors for weight tensor. The authors propose a method called DOMR, which decouples the scale factors of weights at quant/dequant processes, considering the fact that integer weights can be obtained early in the process before actual deployment. (ii) Handling outliers: Most existing methods ignore outliers in model activations that fall outside the clipping range, particularly in lightweight models and low-bit settings. The authors introduce a technique called Outlier-Migrated Reconstruction to save outliers within a pre-defined bitwidth. The experimental results demonstrate that DOMR outperforms existing methods and establishes a new state-of-the-art approach in PTQ. Specifically, it achieves a 12.93% improvement in Top-1 accuracy for the W2A2 configuration on MobileNet-v2. The authors also mention that they will release the code associated with their work, which is rather important considering the complexity of the proposed method.

**Strengths:**

- The paper is exceptionally well-crafted, featuring clear and insightful illustrations that greatly aid readers in comprehending the presented concepts.
- The authors conducts sufficient experiments.
- The authors provide intriguing and valuable insights, particularly in the idea of enhancing both weights and activations quantization. The decoupling of weight step-sizes for optimization is a noteworthy contribution, and the notable performance enhancements achieved by introducing a single additional step-size in each layer is quite interesting.

**Weaknesses:**

- The paper would benefit from improved clarity in its notations. The use of variables such as w_l, w_u, x_l, and x_u in Equation 2 is not well-defined. It's crucial for the authors to provide clear explanations and definitions for these lower and upper bounds of weights and activations.
- The authors base their Outlier-Migrated Reconstruction (OMR) method on the assumption that the activation function is ReLU. However, it's essential to discuss the applicability of OMR to models that use non-ReLU functions. For instance, MobileNetV3 employs h-swish, and ViT utilizes GeLU. A more comprehensive discussion about the adaptability of OMR to such activation functions would enhance the paper's coverage.
- The paper mentions that OMR involves duplicating channels for both weights and activations. This duplication may introduce additional computational overhead, but the paper lacks an in-depth analysis of this aspect. Providing a more detailed examination of the computational costs could be better.
- The inclusion of an overall cost breakdown in Table 11 is appreciated. However, it would be even more informative if the authors could provide a specific breakdown of the runtime costs associated with Quant Time in the Fake Quantization process. This would offer a more granular view of the computational expenses involved in the proposed method, contributing to a deeper understanding of its practical implications.

**Questions:**

It seems that using decoupled step sizes for weights can also generalize to quantization-aware training, have the authors experimented with this?

---

> ### Author Response · Authors · 2023-11-21
> **OMR on other activation function; Runtime Costs in Fake-Quant; DJOS on QAT**
>
> Thanks for your instructive review.
> ## 1.1 Mathematics Notation
> We have added the mathematics notions, like $w_u, w_l,x_u,x_l$ in our re-polished paper PDF.
> ## 1.2 OMR on ReLU, h-swish, GeLU and others.
> Except for Conv-ReLU-Conv and positive nonlinear function like ReLU6/Sigmoid, **the core of OMR, migrating outliers into safe clipping range then compensating in the following layers, can be extented to other structures like Conv-Conv, Linear-Linear and nonlinear function like h-swish, GeLU.**
>
> $x_{int}^{min}(x_{int}^{max}),X_{clip}^{min}(X_{clip}^{max})$ denote the min(max) value(clipping bound) in integer and fp32 domain.  Activation fake quantization into B bits can be obtained by,
> $$
> \hat{x}=s_x\cdot x_{int}=s_x\cdot clip\\{{\lfloor\frac{x}{s_x}\rceil}, x_{int}^{min}, x_{int}^{max}\\}
> $$
> Detailed visualization is shown in **Sec4 of the re-polished supplementary PDF**.
>
> - When activation function is **ReLU**
>     - $x_{int}^{min}=0, x_{int}^{max}=2^B-1$ , $X_{clip}^{min}=0,X_{clip}^{max}=(2^B-1)*s_w$, The inner values is in range $[X_{clip}^{min}, X_{clip}^{max}]$. The outliers is in range $(X_{clip}^{max}, +\infty)$. We want to save outliers in $(X_{clip}^{max},2X_{clip}^{max}]$. Thus we can migrate $x$ by $X_{clip}^{max}-X_{clip}^{min}=(2^B-1)*s_w$ along the negative $x$-axis to save values in  $(X_{clip}^{max},2X_{clip}^{max}]$.
> - When activation function is **GeLU, h-swish** or **no activation function**, activation $x$ distributes in both negative and positive axis:
>     - $x_{int}^{min}=-(2^{B-1}-1), x_{int}^{max}=2^{B-1}-1$ ,$X_{clip}^{min}=-(2^{B-1}-1)*s_w,X_{clip}^{max}=(2^{B-1}-1)*s_w$, The inner values is in range $[X_{clip}^{min}, X_{clip}^{max}]$. The outliers is in range $(-\infty, X_{clip}^{min})$ and  $(X_{clip}^{max}, +\infty)$. We want to save outliers in $[2X_{clip}^{min},X_{clip}^{min})$ and  $(X_{clip}^{max},2X_{clip}^{max}]$. Thus we can migrate $x$ by  $X_{clip}^{max}-X_{clip}^{min}$ along the negative $x$-axis to save values in $(X_{clip}^{max},2X_{clip}^{max}]$. We can migrate $x$ by $X_{clip}^{max}-X_{clip}^{min}$ along the positive $x$-axis to save values in $[2X_{clip}^{min},X_{clip}^{min})$.
>
> ## 1.3 Detail of Computational Costs
> Our OMR does no need the Fake Quant process. It can be finished in **1 CPU-second** based on offline mathematical transformation. It introduces extra FLOPs during Real Quant.
> - For k=0.2 or k=0.5, where we save 20% or 50% channles' outliers, if
> FLOPs of original OMR-Solvalble structures is 10M, then FLOPs of
> OMR-applied ones is 12M or 15M.
> - Trade-off analysis on MobileNet-V2 between FLOPs/BOPs and its performance is as follows. We denote the FLOPs 327M and BOPs 1.8G of MobileNet-V2 on W2A2 as 1.0, on which of other methods is based. **Note** that "W2A3" can not run on 2-bit hardware while our **OMR can run on 2-bit hardware with W2A3 performance** as our paper PDF.
>     | Mobile-V2 | W2A2 | FLOPs | BOPs | Run on 2bit Hardware |
>     |-|-|-|-|-|
>     | NWQ+Ori_Net | 26.42 | 1.0 | 1.0 | √ |
>     | NWQ+Channel-plus -Net | 32.02 | 2.0 | 2.0 | √ |
>     | **OMR_0.0+Ori-Net(DJOS)** | **31.43** | **1.0** | **1.0** | √ |
>     | OMR_0.3+Ori-Net | 36.33 | 1.3 | 1.3 | √ |
>     | **OMR_0.5+Ori-Net** | **39.35** | 1.5  | 1.5 | √ |
>     | OMR_0.7+Ori-Net | 39.75 | 1.7 | 1.7 | √ |
>     | OMR_1.0_Ori-Net | 41.65 | 2.0 | 2.0 | √ |
>
> ## 1.4 Runtime Costs in Fake-Quant
> The specific breakdown of runtime costs of our DOMR(DJOS+OMR) in ResNet-18 fake quantization period is as follows：
>
> 1.4.1. Fold BN into its preceding convolution in FP32 model: 0.02 second.
>
> 1.4.2. Initialize the weight’s single quant-step $s_w$ with MSE minimization, decouple the single $s_w$ into $s_w,s^\prime_w$:  0.5 second
>
> 1.4.3. OMR’s sensitivity analysis on channels + choosing sensitive channels for OMR + weight adjustment of the current layer and the next layer: 0.42 minutes
>
> 1.4.4. Jointly optimize $s_w^\prime,s_x$, AdaRound param of weight $\alpha$ on 1024 random calibration images for 20K iterations: 44 minutes
>
> All the time consumed on our DOMR fake quantization: 0.02s + 0.5s + 0.42 min + 44 min ≈ 44.43 min
> ## 1.5 DJOS on QAT(Quantization-Aware Training)
> | ResNet18 | W3A3 | W2A2 |
> |-|-|-|
> | LSQ_QAT | 69.57±0.08 | 66.50±0.21 |
> | LSQ_QAT+DJOS | 68.95±0.07 | 62.57±0.18 |
>
> We re-implement LSQ-QAT W2A2 and W3A3 on ResNet-18. Experimental results over 3 runs can be seen above. We can see LSQ_QAT+DJOS performs worse than original LSQ_QAT.
>
> The reason is that model weight in QAT is learnable and its distribution changes over training process. Whether to learn model weights is also one of the biggest difference for PTQ and QAT.
>
> Thus a fixed quant-step $s_w$ of DJOS can not be adaptive to a weight distribution that changes over training process. However, in PTQ, model weight is almost fixed, with at most one quant-step changes due to AdaRound. Therefore,  with limited calibration set and limited optimization time in PTQ, a fixed quant-step $s_w$ will perform better on an almost-fixed weight distribution.

---

> ### Author Response · Authors · 2023-11-22
>
> Dear Reviewer RfJv,
>
> Thanks for your valuable reviews.
>
> If you have any problem, we are online for response by the last second of 22nd November. We greatly value your comments and are committed to refining our work.
>
> Thank you for your time and expertise.
>
> Sincerely,
>
> Authors

---

> > ### Comment · Reviewer_RfJv · 2023-11-22
> >
> > Thanks for your reply. While some of my concerns were addressed, I still recommend the authors provide results for non-relu models such as mbv3. At the same time, I partially agree with the comments of the other two reviewers that the difference in high-level idea between the proposed method and OCS is not significant. While this may not be a critical factor for me, I believe it merits thorough discussion in the paper, including the provision of experiments to analyze the quantization noise. I want to keep my rating at borderline acceptance.

---

> ### Author Response · Authors · 2023-11-22
> **OMR is Novel, V.s. OCS**
>
> Dear reviewer  RfJv,
>
> Thanks for your valuable feedback.
>
> We will add experiments for MobileNet-V3 to enhance our paper.
>
> The difference of OMR and OCS is more significant than what review pf4t said before. **The novelty of OMR has been approved by reviewer wiuF just now.**
> >Thanks for the answer to my last question. It helps understand OMR shows better expressibility compared to OCS. Therefore, I decided to raise my score to 6.
>
> Detailed analysis is as follows,
>
> ## 1. OMR V.s. OCS
> **Our OMR is motivated from OCS to overcome OCS's shortcoming of sacrificing the precision of inner values. Our OMR, for the first time, proposes to save more outliers while preserve the precision of inner values.**
>
>
>
> It is true that when integer activation is  [2,4,8,16], OMR share the same precision as OCS as reviewer pf4t.
>
> However, when input is [2,4,8,15], OCS will split split [2,4,8,15]->[1,2,4,7]+[1,2,4,7] or [2,4,8,15]->[1,2,4,8]+[1,2,4,8], thus there will be **an rounding error** for 15->7\*2=14 or 15->8\*2=16, while OMR will split [2,4,8,15]->[1,2,4,8]+[1,2,4,7], thus no error is caused for 15->8+7.
>
> The main problem is that when the bin's number of inputs is larger than quantization levels, OCS saves outliers by sacrificing the precision of inner values.
>
> For a more detailed example with the whole fake quantization: x=[0.0, 0.1, 0.2, 0.3, *0.4*, *0.5*, *0.6*, *0.7*], step=0.1, 2-bit, Thus outliers is [0.4,0.5,0.6,0.7].
> - Normal 2-bit:
>   - $x_{int}=clip[round(x/step), 0, 3]=[0, 1, 2, 3, 3, 3, 3, 3]$ quantization levels: 4
>   - $\hat{x}=x_{int}\*step=[0.0, 0.1, 0.2, 0.3, 0.3, 0.3, 0.3, 0.3]$ quantization levels: 4
> - OCS 2-bit:
>   - $x_{int}=clip[round((x/2)/step), 0, 3]=[0, 0, 1, 1, 2, 2, 3, 3]$ quantization levels: 4
>   - $\hat{x}=(x_{int}\*step)*2=[0.0, 0.0, 0.2,0.2,  0.4, 0.4, 0.6,0.6]$ quantization levels: 4
> - OMR 2-bit:
>   - $x_{int}=concat\\{clip[round((x)/step), 0, 3], clip[round((x-0.4)/step), 0, 3]\\}=[[0, 1,  2, 3],[0,1,2,3]]$
>   - $\hat{x}=concat\\{x_{int}[0]\*step, x_{int}[1]*step+0.4]\\}=concat\\{[0.0, 0.1, 0.2, 0.3], [0.4,0.5, 0.6, 0.7]\\}=[0.0, 0.1, 0.2, 0.3, 0.4, 0.5, 0.6, 0.7]$ **quantization levels: 8**
>
> Therefore:
> - **OMR is closer to (here equal to) original fp32 values during fake-quant with larger quantization levels**.
>   - [0.0, 0.1, 0.2, 0.3, 0.4, 0.5, 0.6, 0.7] $\rightarrow$ [0.0, 0.1, 0.2, 0.3, 0.4, 0.5, 0.6, 0.7]
> - OCS sacrifices [0.0, 0.1, 0.2, 0.3, 0.4, 0.5, 0.6, 0.7] $\rightarrow$  [0.0, 0.0, 0.2,0.2,  0.4, 0.4, 0.6,0.6] during fake-quant.
>
> In conclusion:
> - OCS does not enlarge quantization levels. OCS saves outliers while sacrifices the precision of inner values.
> - Our OMR, for the first time, proposes to save more outliers while preserve the precision of inner values.
>
> **Comparison visualization is re-polished in supplementary PDF Sec2.**
>
> If you have any problem, we are online for response by the last second of 22nd November. We are reaching out to see if our response adequately addresses your concerns. We greatly value your comments and are committed to refining our work.
>
> Sincerely,
>
> authors

---

### Meta-Review · Area_Chair_Hxrv · 2023-12-10

**Metareview:**

The submission presents an approach called DOMR for improving the performance of post-training quantization (PTQ). DOMR combines two techniques called DJOS and OMR: DJOS decouples the quant step into a quant step and a dequant step and only optimizes the dequant step, and OMR saves outliers with a predefined bitwidth. The proposed approach is evaluated on the ImageNet and COCO tasks and is shown to outperform competing approaches.
Reviewers note the submission's overall clarity and writing quality (with the exception of its mathematical notation; RfJv, wiuF) and its extensive experimental support (RfJv, pf4t, wiuF). Reviewers pf4t and wiuF also note the approach is straightforward and effective, and Reviewer RfJv notes the submission provides valuable insights. Reviewer concerns regarding mathematical notation clarity (RfJv), the lack of a discussion on the computational overhead of the proposed approach (RfJv, wiuF), the fairness of the evaluation against competing approaches (pf4t), and the isolated effect of DJOS (pf4t) have been addressed by the authors in their response.

The two main outstanding reviewer concerns are:

- Reviewer RfJv wonders how generalizable the proposed approach is beyond network architectures with ReLU activations. The authors provide an explanation of how the approach can be extended to other nonlinearities, and the reviewer maintains their recommendation to provide results for non-relu models such as MobileNet-v3, which the authors promise to do (but have not done yet as far as I can tell).
- Through their reviews and discussions with the authors, all three reviewers expressed the concern that OMR is very similar to OCS and therefore lacks technical novelty. The authors explains that OCS saves outliers while sacrificing the precision of inner values, whereas OMR preserves the precision of inner values and refer reviewers to the supplementary material for details. Reviewer wiuF appears satisfied with the explanation and raises their score to a 6 as a result.

Ultimately the submission is a borderline case, and while reviewers recognize merits in the paper, no reviewer is strongly in favor of acceptance and willing to champion the submission. As a result, it does not quite meet the bar for acceptance.

**Justification For Why Not Higher Score:**

While reviewers recognize merits in the paper, no reviewer is strongly in favor of acceptance and willing to champion the submission. As a result, it does not quite meet the bar for acceptance.

**Justification For Why Not Lower Score:**

N/A

---

### Decision · Program_Chairs · 2024-01-16

Reject